# ONLINE CORESET SELECTION FOR REHEARSAL-BASED CONTINUAL LEARNING

**Jaehong Yoon**[1]   **Divyam Madaan**[2*]   **Eunho Yang**[1,3]   **Sung Ju Hwang**[1,3]
KAIST[1]   New York University[2]   AITRICS[3]
jaehong.yoon@kaist.ac.kr,   divyam.madaan@nyu.edu,
eunhoy@kaist.ac.kr,   sjhwang82@kaist.ac.kr

## ABSTRACT

A dataset is a shred of crucial evidence to describe a task. However, each data point in the dataset does not have the same potential, as some of the data points can be more representative or informative than others. This unequal importance among the data points may have a large impact in rehearsal-based continual learning, where we store a subset of the training examples (coreset) to be replayed later to alleviate catastrophic forgetting. In continual learning, the quality of the samples stored in the coreset directly affects the model's effectiveness and efficiency. The coreset selection problem becomes even more important under realistic settings, such as imbalanced continual learning or noisy data scenarios. To tackle this problem, we propose *Online Coreset Selection (OCS)*, a simple yet effective method that selects the most representative and informative coreset at each iteration and trains them in an online manner. Our proposed method maximizes the model's adaptation to a current dataset while selecting high-affinity samples to past tasks, which directly inhibits catastrophic forgetting. We validate the effectiveness of our coreset selection mechanism over various standard, imbalanced, and noisy datasets against strong continual learning baselines, demonstrating that it improves task adaptation and prevents catastrophic forgetting in a sample-efficient manner.

## 1 INTRODUCTION

Humans possess the ability to learn a large number of tasks by accumulating knowledge and skills over time. Building a system resembling human learning abilities is a deep-rooted desire since sustainable learning over a long-term period is essential for general artificial intelligence. In light of this need, *continual learning* (CL) (Thrun, 1995), or *lifelong learning*, tackles a learning scenario where a model continuously learns over a sequence of tasks (Kumar & Daume III, 2012; Li & Hoiem, 2016) within a broad research area, such as classification (Kirkpatrick et al., 2017; Chaudhry et al., 2019a), image generation (Zhai et al., 2019), language learning (Li et al., 2019b; Biesialska et al., 2020), clinical application (Lee & Lee, 2020; Lenga et al., 2020), speech recognition (Sadhu & Hermansky, 2020), and federated learning (Yoon et al., 2021). A well-known challenge for continual learning is *catastrophic forgetting* (McCloskey & Cohen, 1989), where the continual learner loses the fidelity for past tasks after adapting the previously learned knowledge to future tasks.

Recent rehearsal-based continual learning methods adapt the continual model to the previous tasks by maintaining and revisiting a small replay buffer (Titsias et al., 2020; Mirzadeh et al., 2020). However, the majority of these methods store random-sampled instances as a proxy set to mitigate catastrophic forgetting, limiting their practicality to real-world applications (see Figure 1a) when all the training instances are not equally useful, as some of them can be more representative or informative for the current task, and others can lead to performance degeneration for previous tasks. Furthermore, these unequal potentials could be more severe under practical scenarios containing *imbalanced, streaming, or noisy instances* (see Figure 2). This leads to an essential question in continual learning:

> *How can we obtain a coreset to promote task adaptation for the current task while minimizing catastrophic forgetting on previously seen tasks?*

---

[*]The work was done while the author was a student at KAIST.

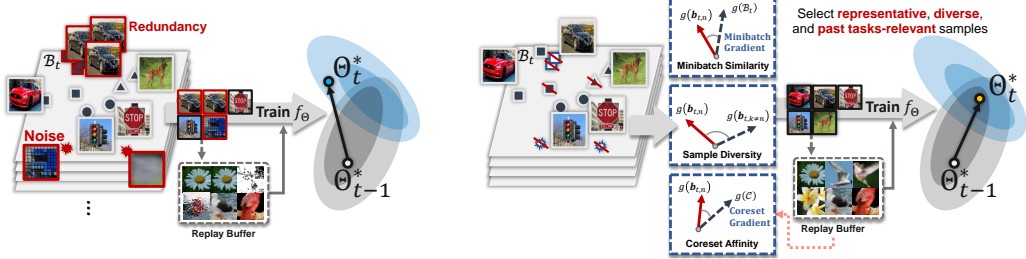

(a) Existing rehearsal-based CL  (b) Online Coreset Selection (OCS)

Figure 1: **Illustration of existing rehearsal-based CL and Online Coreset Selection (OCS): (a)** Existing rehearsal-based methods train on all the arrived instances and memorize a fraction of them in the replay buffer, which results in a suboptimal performance due to the outliers (noisy or biased instances). **(b)** OCS obtains the coreset by leveraging our three selection strategies, which discard the outliers at each iteration. Consequently, the selected examples promote generalization and minimize interference with the previous tasks.

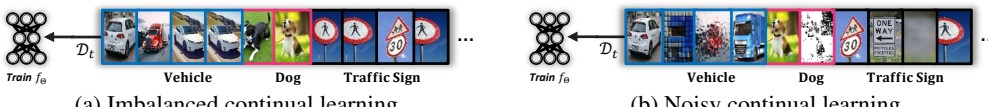

(a) Imbalanced continual learning  (b) Noisy continual learning

Figure 2: **Realistic continual learning scenarios: (a)** Each task consists of class-imbalanced instances. **(b)** Each task has uninformative noise instances, which hamper training.

To address this question, we propose *Online Coreset Selection (OCS)*, a novel method for continual learning that selects representative training instances for the current and previous tasks from arriving streaming data in an online fashion based on our following three selection strategies: **(1)** *Minibatch similarity* selects samples that are representative to the current task $\mathcal{T}_t$. **(2)** *sample diversity* encourages minimal redundancy among the samples of current task $\mathcal{T}_t$. **(3)** *Coreset affinity* promotes minimum interference between the selected samples and knowledge of the previous tasks $\mathcal{T}_k$, $\forall k < t$. To this end, OCS minimizes the catastrophic forgetting on the previous tasks by utilizing the obtained coreset for future training, and also encourages the current task adaptation by updating the model parameters on the top-$\kappa$ selected data instances. The overall concept is illustrated in Figure 1b.

Our method is simple, intuitive, and is generally applicable to any rehearsal-based continual learning method. We evaluate the performance of OCS on various continual learning scenarios and show that it outperforms state-of-the-art rehearsal-based techniques on balanced, imbalanced, and noisy continual learning benchmarks of varying complexity. We also show that OCS is general and exhibits collaborative learning with the existing rehearsal-based methods, leading to increased task adaptation and inhibiting catastrophic forgetting. To summarize, our contributions are threefold:

- We address the problem of coreset selection for realistic and challenging continual learning scenarios, where the data continuum is composed of class-imbalanced or noisy instances that deteriorate the performance of the continual learner during training.

- We propose *Online Coreset Selection (OCS)*, a simple yet effective online coreset selection method to obtain a representative and diverse subset that has a high affinity to the previous tasks from each minibatch during continual learning. Specifically, we present three gradient-based selection criteria to select the coreset for current task adaptation while mitigating catastrophic forgetting.

- We demonstrate that OCS is applicable to any rehearsal-based continual learning method and experimentally validate it on multiple benchmark scenarios, where it largely improves the performance of the base algorithms across various performance metrics.

## 2  RELATED WORK

**Continual learning.** In the past few years, there has been significant progress in continual learning to alleviate catastrophic forgetting (McCloskey & Cohen, 1989). The *regularization approaches* (Kirkpatrick et al., 2017; Lee et al., 2017; Serrà et al., 2018) modify the model parameters with additional regularization constraints to prevent catastrophic forgetting. The *architecture approaches* (Rusu et al., 2016; Yoon et al., 2018; Xu & Zhu, 2018; Li et al., 2019a; Yoon et al., 2020) utilize network isolation or expansion during continual learning to improve network performance. Another line of research uses *rehearsal approaches*, which memorize or generate a small fraction of data points for previous tasks and utilizes them to retain the task knowledge (Lopez-Paz & Ranzato, 2017; Chaudhry et al., 2019a; Aljundi et al., 2019b; Borsos et al., 2020). For example, Gradient-based Sample Selection

(GSS) (Aljundi et al., 2019b) formulates the selection of the replay buffer as a constraint selection problem to maximize the variance of gradient direction. ER-MIR (Aljundi et al., 2019a) iteratively constructs the replay buffer using a loss-based criterion, where the model selects the top-$\kappa$ instances that increase the loss between the current and previous iteration. However, the existing rehearsal-based methods (Rebuffi et al., 2017; Aljundi et al., 2019b;a; Chaudhry et al., 2019a;b) do not select the coreset before the current task adaptation and update the model on all the arriving data streams, which makes them susceptible to real-world applications that include noisy and imbalanced data distributions. In contrast, OCS selects the instances before updating the model using our proposed selection criteria, which makes it robust to *past and current task training* across various CL scenarios.

**Coreset selection.** There exist various directions to obtain a coreset from a large dataset. Importance sampling (Johnson & Guestrin, 2018; Katharopoulos & Fleuret, 2018; Sinha et al., 2020) strengthens the loss/gradients of important samples based on influence functions. Kool et al. (2019) connect stochastic Gumbel-top-$k$ trick and beam search to hierarchically sample sequences without replacement. Rebuffi et al. (2017) propose a herding based strategy for coreset selection. Nguyen et al. (2018) formulate the coreset summarization in continual learning using online variational inference (Sato, 2001; Broderick et al., 2013). Aljundi et al. (2019b) select the replay buffer to maximize the variance in the gradient-space. Contrary to these methods, OCS considers the diversity, task informativity and relevancy to the past tasks. Recently, Borsos et al. (2020) propose a bilevel optimization framework with cardinality constraints for coreset selection. However, their method is extremely limited in practice and inapplicable in large-scale settings due to the excessive computational cost incurred during training. In contrast, our method is simple, and scalable since it can construct the coreset in the online streaming data continuum without additional optimization constraints.

## 3 REHEARSAL-BASED CONTINUAL LEARNING

We consider learning a model over a sequence of tasks $\{\mathcal{T}_1, \ldots, \mathcal{T}_T\} = \mathcal{T}$, where each task is composed of independently and identically distributed datapoints and their labels, such that task $\mathcal{T}_t$ includes $\mathcal{D}_t = \{\boldsymbol{x}_{t,n}, y_{t,n}\}_{n=1}^{N_t} \sim \mathcal{X}_t \times \mathcal{Y}_t$, where $N_t$ is the total number of data instances, and $\mathcal{X}_t \times \mathcal{Y}_t$ is an unknown data generating distribution. We assume that an arbitrary set of labels for task $\mathcal{T}_t$, $\mathbf{y}_t = \{y_{t,n}\}_{n=1}^{N_t}$ has unique classes, $\mathbf{y}_t \cap \mathbf{y}_k = \emptyset, \forall t \neq k$. In a standard continual learning scenario, the model learns a corresponding task at each step and $t$-th task is accessible at step $t$ only. Let neural network $f_\Theta : \mathcal{X}_{1:T} \to \mathcal{Y}_{1:T}$ be parameterized by a set of weights $\Theta = \{\theta_l\}_{l=1}^L$, where $L$ is the number of layers in the neural network. We define the training objective at step $t$ as follows:

$$\underset{\Theta}{\text{minimize}} \sum_{n=1}^{N_t} \ell(f_\Theta(\boldsymbol{x}_{t,n}), y_{t,n}), \tag{1}$$

where $\ell(\cdot)$ is any standard loss function (e.g., cross-entropy loss). The naive CL design where a simple sequential training on multiple tasks without any means for tackling catastrophic forgetting cannot retain the knowledge of previous tasks and thus results in catastrophic forgetting. To tackle this problem, rehearsal-based methods (Nguyen et al., 2018; Chaudhry et al., 2019a; Titsias et al., 2020) update the model on a randomly sampled replay buffer $\mathcal{C}_k$ constructed from the previously observed tasks, where $\mathcal{C}_k = \{\boldsymbol{x}_{k,j}, y_{k,j}\}_{j=1}^{J_k} \sim \mathcal{D}_k, \forall k < t$ and $J_k \ll N_k$. Consequently, the quality of selected instances is essential for rehearsal-based continual learning. For example, some data instances can be more informative and representative than others to describe a task and improve model performance. In contrast, some data instances can degrade the model's memorization of past tasks' knowledge. Therefore, obtaining the most beneficial examples for the current task is crucial for the success of rehearsal-based CL methods.

To validate our hypothesis, we design a learning scenario with a sequence of two tasks, MNIST ($\mathcal{T}_1$) → CIFAR-10 ($\mathcal{T}_2$) using ResNet-18. After the standard single epoch training on $\mathcal{T}_1$, we update the model weights through a single back-propagation step using a randomly selected data point from $\mathcal{T}_2$, and measure test accuracy of its corresponding class $c$ and forgetting of the entire dataset of a past task $\mathcal{T}_1$. Results for individual impacts on 1000 data points from $\mathcal{T}_2$ are

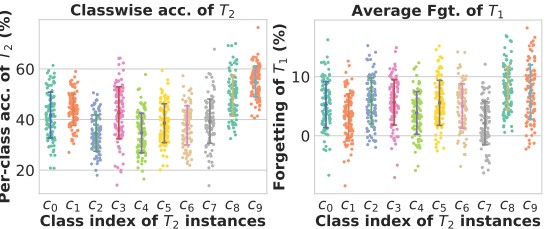

Figure 3: **Per-class accuracy and average forgetting** when a model trained on MNIST ($\mathcal{T}_1$) is updated on a single data point at class $c$ on CIFAR-10 ($\mathcal{T}_2$).

described in Section 3. The influence of each data point from $\mathcal{T}_2$ has a large disparity not only on the corresponding class accuracy but also on past task's forgetting that results in a very high standard deviation. We emphasize that each data point has a different potential impact in terms of forgetting past tasks. Few data points are much more robust to catastrophic forgetting than others, and this can be severe when the influences are accumulated during training.

Based on this motivation, our objective is to select the data instances that can promote current task adaptation while minimizing catastrophic forgetting on the previous tasks. We propose a selection criterion that selects the subset that maximizes the gradient similarity between the representative instances and the current task dataset. More formally:

$$\mathbf{u}^* = \underset{\mathbf{u} \in \mathbb{N}^\kappa}{\text{maximize}}\, \mathcal{S}\left(\frac{1}{N_t} \nabla f_\Theta\left(\mathcal{D}_t\right), \frac{1}{\kappa}\sum_{n \in \mathbf{u}} \nabla f_\Theta\left(\boldsymbol{x}_{t,n}, y_{t,n}\right)\right), \quad \text{where } \mathbf{u} = \{n : n \in \mathbb{N}_{<N_t}\}, \quad (2)$$

where $\mathcal{S}$ is any arbitrary similarity function and $\mathbf{u}^*$ is an index set that selects top-$\kappa$ informative samples without replacement. However, obtaining a representative subset from the entire dataset is computationally expensive and intractable for online continual learning; therefore, we consider a minibatch as an approximation of the dataset and select few representative data instances at each minibatch iteration. We empirically validate that our approximation generally holds across various datasets, network structures, and minibatch sizes in Appendix B and Figure B.9. Consequently, the model iteratively updates the parameters to find the optimal local minima of the loss using informative data points, which obtain similar gradient directions with the averaged gradients of the dataset. In the next section, we propose *OCS* which consists of a simple similarity criterion to achieve this objective. However, similarity criterion is not sufficient to select the representative coreset for online continual learning; hence, we propose diversity and coreset affinity criteria to mitigate catastrophic forgetting.

## 4 ONLINE CORESET SELECTION

In this section, we introduce our selection strategies and propose *Online Coreset Selection (OCS)* to strengthen current task adaptation and mitigate catastrophic forgetting. Thus far, the rehearsal-based continual learning methods (Rebuffi et al., 2017; Aljundi et al., 2019b;a; Chaudhry et al., 2019a;b) populate the replay buffer to preserve the knowledge on the previous tasks. However, we argue that some instances may be non-informative and inappropriate to construct the replay buffer under realistic setups (such as video streaming or imbalanced continual learning scenarios), leading to the degradation of the model's performance. Moreover, it is critical to select the valuable samples for current task training since the model can easily overfit to the biased and noisy data stream, which negatively affects the model generalization. To satisfy these desiderata, we propose *minibatch similarity* ($\mathcal{S}$) and *sample diversity* ($\mathcal{V}$) criteria based on our aforementioned assumption to adaptively select the useful instances without the influence of outliers.

**Definition 1 (Minibatch similarity).** *Let $\boldsymbol{b}_{t,n} = \{\boldsymbol{x}_{t,n}, y_{t,n}\} \in \mathcal{B}_t$ denote $n$-th pair of data point with gradient $\nabla f_\Theta\left(\boldsymbol{b}_{t,n}\right)$ and its corresponding label at task $\mathcal{T}_t$. Let $\bar{\nabla} f_\Theta(\mathcal{B}_t)$ denote the averaged gradient vector of $\mathcal{B}_t$. The minibatch similarity $\mathcal{S}\left(\boldsymbol{b}_{t,n} \mid \mathcal{B}_t\right)$ between $\boldsymbol{b}_{t,n}$ and $\mathcal{B}_t$ is given by*

$$\mathcal{S}\left(\boldsymbol{b}_{t,n} \mid \mathcal{B}_t\right) = \frac{\nabla f_\Theta\left(\boldsymbol{b}_{t,n}\right) \bar{\nabla} f_\Theta\left(\mathcal{B}_t\right)^\top}{\|\nabla f_\Theta\left(\boldsymbol{b}_{t,n}\right)\| \cdot \|\bar{\nabla} f_\Theta\left(\mathcal{B}_t\right)\|}. \quad (3)$$

**Definition 2 (Sample diversity).** *Let $\boldsymbol{b}_{t,n} = \{\boldsymbol{x}_{t,n}, y_{t,n}\} \in \mathcal{B}_t$ denote $n$-th pair of a data point with gradient $\nabla f_\Theta\left(\boldsymbol{b}_{t,n}\right)$ and its corresponding label at task $\mathcal{T}_t$. The sample diversity $\mathcal{V}\left(\boldsymbol{b}_{t,n} \mid \mathcal{B}_{t\setminus\boldsymbol{b}_{t,n}}\right)$ between $\boldsymbol{b}_{t,n}$ and all other instances in $\mathcal{B}_t$ $\left(\mathcal{B}_{t\setminus\boldsymbol{b}_{t,n}}\right)$ is given by*

$$\mathcal{V}\left(\boldsymbol{b}_{t,n} \mid \mathcal{B}_{t\setminus\boldsymbol{b}_{t,n}}\right) = \frac{-1}{N_t - 1}\sum_{p \neq n}^{N_t - 1} \frac{\nabla f_\Theta\left(\boldsymbol{b}_{t,n}\right) \nabla f_\Theta\left(\boldsymbol{b}_{t,p}\right)^\top}{\|\nabla f_\Theta\left(\boldsymbol{b}_{t,n}\right)\| \cdot \|\nabla f_\Theta\left(\boldsymbol{b}_{t,p}\right)\|}. \quad (4)$$

In particular, minibatch similarity considers a minibatch as an approximation of the current task dataset and compares the minibatch-level similarity between the gradient vector of a data point $\boldsymbol{b}$ and its minibatch $\mathcal{B}$. It measures how well a given data instance describes the current task at each training step. Note that selecting examples with the largest minibatch similarity is reasonable when the variance of task instances is low; otherwise, it increases the redundancy among coreset items. In

contrast, we formulate the sample diversity of each data point $b \in \mathcal{B}$ as an averaged dissimilarity (i.e., an average of negative similarities) between a data point itself and other samples in the same minibatch $\mathcal{B}$, and not as an average similarity. Thus, the measure of sample diversity in Equation (4) is negative and the range is $[-1, 0]$.

## 4.1 Online Coreset Selection for Current Task Adaptation

The model receives a data continuum during training, including noisy or redundant data instances in real-world scenarios. Consequently, the arriving data instances can interrupt and hurt the performance of the model. To tackle this problem, we consider an amalgamation of minibatch similarity and sample diversity to select the most helpful instances for current task training. More formally, our online coreset selection for the current task adaptation can be defined as follows:

$$\mathbf{u}^* = \left\{ \underset{n}{\operatorname{argmax}}^{(\kappa)} \, \mathcal{S}\left(\boldsymbol{b}_{t,n} \mid \mathcal{B}_t\right) + \mathcal{V}\left(\boldsymbol{b}_{t,n} \mid \mathcal{B}_{t \setminus \boldsymbol{b}_{t,n}}\right) \; \middle| \; n \in \{0, \ldots, |\mathcal{B}_t| - 1\} \right\}. \tag{5}$$

We emphasize that we can obtain the top-$\kappa$ valuable instances for the current task by computing Equation 5 in an online manner. Once the representative coreset is selected, we optimize the following objective for the current task training at each iteration:

$$\underset{\Theta}{\operatorname{minimize}} \; \frac{1}{\kappa} \sum_{(\hat{\boldsymbol{x}}, \hat{y}) \in \widehat{\mathcal{B}}_t}^{\kappa} \ell\left(f_\Theta\left(\hat{\boldsymbol{x}}\right), \hat{y}\right), \quad \text{where } \widehat{\mathcal{B}}_t = \mathcal{B}_t[\mathbf{u}^*]. \tag{6}$$

We consider a selected coreset at each iteration as a candidate for the replay buffer. After the completion of each task training, we choose a coreset $\mathcal{C}_t$ among the collected candidates, or we may also iteratively update $\mathcal{C}_t$ to maintain the bounded buffer size for continual learning.

## 4.2 Online Coreset Selection for Continual Learning

We now formulate OCS for online continual learning, where our objective is to obtain the coreset to retain the knowledge of the previous tasks using our proposed similarity and diversity selection criteria. However, continual learning is more challenging as the model suffers from catastrophic forgetting and coreset size is smaller than the size of the arriving data streams. Thus, inspired by our observation in Section 3, we aim to train the continual learner on the selected instances that are representative of the current task and prevent the performance degeneration of previous tasks.

We achieve our goal by introducing our Coreset affinity criterion $\mathcal{A}$ to Equation 5. In particular, $\mathcal{A}$ computes the gradient vector similarity between a training sample and the coreset for previous tasks ($\mathcal{C}$). More formally, $\mathcal{A}$ can be defined as follows:

**Definition 3 (Coreset affinity).** *Let $\boldsymbol{b}_{t,n} = \{\boldsymbol{x}_{t,n}, y_{t,n}\} \in \mathcal{B}_t$ denote the $n$-th pair of a data point with gradient $\nabla f_\Theta\left(\boldsymbol{b}_{t,n}\right)$ and its corresponding label at task $\mathcal{T}_t$. Further let $\bar{\nabla} f_\Theta(\mathcal{B}_\mathcal{C})$ be the averaged gradient vector of $\mathcal{B}_\mathcal{C}$, which is randomly sampled from the coreset $\mathcal{C}$. The coreset affinity $\mathcal{A}\left(\boldsymbol{b}_{t,n} \mid \mathcal{B}_\mathcal{C} \sim \mathcal{C}\right)$ between $\boldsymbol{b}_{t,n}$ and $\mathcal{B}_\mathcal{C}$ is given by*

$$\mathcal{A}\left(\boldsymbol{b}_{t,n} \mid \mathcal{B}_\mathcal{C} \sim \mathcal{C}\right) = \frac{\nabla f_\Theta\left(\boldsymbol{b}_{t,n}\right) \bar{\nabla} f_\Theta\left(\mathcal{B}_\mathcal{C}\right)^\top}{\|\nabla f_\Theta\left(\boldsymbol{b}_{t,n}\right)\| \cdot \|\bar{\nabla} f_\Theta\left(\mathcal{B}_\mathcal{C}\right)\|}. \tag{7}$$

While the past task is inaccessible after the completion of its training, our selectively populated replay buffer can be effectively used to describe the knowledge of the previous tasks. The key idea is to select the examples that minimize the angle between the gradient vector of the coreset containing previous task examples and the current task examples. Instead of randomly replacing the candidates in the coreset (Lopez-Paz & Ranzato, 2017; Chaudhry et al., 2019a; Aljundi et al., 2019b; Borsos et al., 2020), $\mathcal{A}$ promotes the selection of examples that do not degenerate the model performance on previous tasks. To this end, we select the most beneficial training instances which are representative and diverse for current task adaptation while maintaining the knowledge of past tasks. In summary, our OCS for training task $\mathcal{T}_t$ during CL can be formulated as:

$$\mathbf{u}^* = \left\{ \underset{n}{\operatorname{argmax}}^{(\kappa)} \, \mathcal{S}\left(\boldsymbol{b}_{t,n} \mid \mathcal{B}_t\right) + \mathcal{V}\left(\boldsymbol{b}_{t,n} \mid \mathcal{B}_{t \setminus \boldsymbol{b}_{t,n}}\right) + \tau \mathcal{A}\left(\boldsymbol{b}_{t,n} \mid \mathcal{B}_\mathcal{C}\right) \; \middle| \; n \in \{0, \ldots, |\mathcal{B}_t| - 1\} \right\}. \tag{8}$$

$\tau$ is a hyperparameter that controls the degree of model plasticity and stability. Note that, during the first task training, we do not have the interference from previous tasks and we select the top-$\kappa$ instances that maximize the minibatch similarity and sample diversity. Given the obtained coreset $\widehat{\mathcal{B}}_t = \mathcal{B}_t[\mathbf{u}^*]$, our optimization objective reflecting the coreset $\mathcal{C}$, is as follows:

$$\underset{\Theta}{\text{minimize}} \quad \frac{1}{\kappa} \sum_{(\hat{\boldsymbol{x}}, \hat{y}) \in \widehat{\mathcal{B}}_t} \ell(f_\Theta(\hat{\boldsymbol{x}}), \hat{y}) + \frac{\lambda}{|\mathcal{B}_\mathcal{C}|} \sum_{(\boldsymbol{x}, y) \in \mathcal{B}_\mathcal{C}} \ell(f_\Theta(\boldsymbol{x}), y), \tag{9}$$

where $\mathcal{B}_\mathcal{C}$ is a randomly sampled minibatch from the coreset $\mathcal{C}$ and $\lambda$ is a hyperparameter to balance the adaptation between the current task and past task coreset. Overall training procedure for *Online Coreset Selection (OCS)* is described in Algorithm 1. To the best of our knowledge, this is the first work that utilizes *selective online training* for the current task training and incorporates the relationship between the selected coreset and the current task instances to promote current task adaptation while minimizing the interference with previous tasks.

---

**Algorithm 1** Online Coreset Selection (OCS)

---

**input** Dataset $\{\mathcal{D}_t\}_{t=1}^T$, neural network $f_\Theta$, hyperparameters $\lambda, \tau$, replay buffer $\mathcal{C} \leftarrow \{\}$, buffer size bound J.
1: **for** task $\mathcal{T}_t = \mathcal{T}_1, \ldots, \mathcal{T}_T$ **do**
2:      $\mathcal{C}_t \leftarrow \{\}$            ▷ Initialize coreset for current task
3:      **for** batch $\mathcal{B}_t \sim \mathcal{D}_t$ **do**
4:          $\mathcal{B}_\mathcal{C} \leftarrow \text{SAMPLE}(\mathcal{C})$       ▷ Randomly sample a batch from the replay buffer
5:          $\mathbf{u}^* = \underset{n \in \{0, \ldots, |\mathcal{B}_t|-1\}}{\text{argmax}^{(\kappa)}} \mathcal{S}\left(\boldsymbol{b}_{t,n} \mid \mathcal{B}_t\right) + \mathcal{V}\left(\boldsymbol{b}_{t,n} \mid \mathcal{B}_{t \setminus b_{t,n}}\right) + \tau \mathcal{A}\left(\boldsymbol{b}_{t,n} \mid \mathcal{B}_\mathcal{C}\right)$    ▷ Coreset selection
6:          $\Theta \leftarrow \Theta - \eta \nabla f_\Theta(\mathcal{B}_t[\mathbf{u}^*] \cup \mathcal{B}_\mathcal{C})$ with Equation (9)     ▷ Model update with selected instances
7:          $\mathcal{C}_t \leftarrow \mathcal{C}_t \cup \widehat{\mathcal{B}}_t$
8:      **end for**
9:      $\mathcal{C} \leftarrow \mathcal{C} \cup \text{SELECT}(\mathcal{C}_t, \text{size} = J/T)$ with Equation (8)     ▷ Memorize coreset in the replay buffer
10: **end for**

---

## 5 EXPERIMENTS

### 5.1 EXPERIMENTAL SETUP

**Datasets.** We validate OCS on domain-incremental CL for Balanced and Imbalanced Rotated MNIST using a single-head two-layer MLP with 256 ReLU units in each layer, task-incremental CL for Split CIFAR-100 and Multiple Datasets (a sequence of five datasets) with a multi-head structured ResNet-18 following prior works (Chaudhry et al., 2019a; Mirzadeh et al., 2020; 2021). Additionally, we evaluate on class-incremental CL for Balanced and Imbalanced Split CIFAR-100 with a single-head structured ResNet-18 in Table B.9. We perform five independent runs for all the experiments and provide further details on the experimental settings and datasets in Appendix A.

**Baselines.** We compare OCS with regularization-based CL methods: EWC (Kirkpatrick et al., 2017) and Stable SGD (Mirzadeh et al., 2020), rehearsal-based CL methods using random replay buffer: A-GEM (Chaudhry et al., 2019a) and ER-Reservior (Chaudhry et al., 2019b), coreset-based methods using CL algorithms: Uniform Sampling, $k$-means Features (Nguyen et al., 2018) and $k$-means Embeddings (Sener & Savarese, 2018), and coreset-based CL methods: iCaRL (Rebuffi et al., 2017), Grad Matching (Campbell & Broderick, 2019), GSS (Aljundi et al., 2019b), ER-MIR (Aljundi et al., 2019a), and Bilevel Optim (Borsos et al., 2020). We limit the buffer size for the rehearsal-based methods to one example per class per task. Additionally, we compare with Finetune, a naive CL method learnt on a sequence of tasks, and Multitask, where the model is trained on the complete data.

**Metrics.** We evaluate all the methods on two metrics following the CL literature (Chaudhry et al., 2019a; Mirzadeh et al., 2021).
1. **Average Accuracy** $(A_t)$ is the averaged test accuracy of all tasks after the completion of CL at task $\mathcal{T}_t$. That is, $A_t = \frac{1}{t} \sum_{i=1}^t a_{t,i}$, where $a_{t,i}$ is the test accuracy of task $\mathcal{T}_i$ after learning task $\mathcal{T}_t$.
2. **Average Forgetting** $(F)$ is the averaged disparity between the peak and final task accuracy after the completion of continual learning. That is, $F = \frac{1}{T-1} \sum_{i=1}^{T-1} \max_{t \in \{1, \ldots, T-1\}} (a_{t,i} - a_{T,i})$.

Table 1: Performance comparison of OCS and other baselines on balanced and imbalanced continual learning. We report the mean and standard-deviation of the average accuracy (Accuracy) and average forgetting (Forgetting) across five independent runs. The best results are highlighted in **bold**.

| Method | Rotated MNIST | | Split CIFAR-100 | | Multiple Datasets | |
|---|---|---|---|---|---|---|
| | Accuracy | Forgetting | Accuracy | Forgetting | Accuracy | Forgetting |
| **Balanced CL** | | | | | | |
| Finetune | 46.3 ($\pm$ 1.37) | 0.52 ($\pm$ 0.01) | 40.4 ($\pm$ 2.83) | 0.31 ($\pm$ 0.02) | 49.8 ($\pm$ 2.14) | 0.23 ($\pm$ 0.03) |
| EWC (Kirkpatrick et al., 2017) | 70.7 ($\pm$ 1.74) | 0.23 ($\pm$ 0.01) | 48.5 ($\pm$ 1.24) | 0.48 ($\pm$ 0.01) | 42.7 ($\pm$ 1.89) | 0.28 ($\pm$ 0.03) |
| Stable SGD (Mirzadeh et al., 2020) | 70.8 ($\pm$ 0.78) | 0.10 ($\pm$ 0.02) | 57.4 ($\pm$ 0.91) | 0.07 ($\pm$ 0.01) | 53.4 ($\pm$ 2.66) | 0.16 ($\pm$ 0.03) |
| A-GEM (Chaudhry et al., 2019a) | 55.3 ($\pm$ 1.47) | 0.42 ($\pm$ 0.01) | 50.7 ($\pm$ 2.32) | 0.19 ($\pm$ 0.04) | – | – |
| ER-Reservoir (Chaudhry et al., 2019b) | 69.2 ($\pm$ 1.10) | 0.21 ($\pm$ 0.01) | 46.9 ($\pm$ 0.76) | 0.21 ($\pm$ 0.03) | – | – |
| Uniform Sampling | 79.9 ($\pm$ 1.32) | 0.14 ($\pm$ 0.01) | 58.8 ($\pm$ 0.89) | 0.05 ($\pm$ 0.01) | 56.0 ($\pm$ 2.40) | 0.11 ($\pm$ 0.02) |
| iCaRL (Rebuffi et al., 2017) | 80.7 ($\pm$ 0.44) | 0.13 ($\pm$ 0.00) | 60.3 ($\pm$ 0.91) | 0.04 ($\pm$ 0.00) | 59.4 ($\pm$ 1.43) | 0.07 ($\pm$ 0.02) |
| $k$-means Features (Nguyen et al., 2018) | 79.1 ($\pm$ 1.50) | 0.14 ($\pm$ 0.01) | 59.3 ($\pm$ 1.21) | 0.06 ($\pm$ 0.01) | 53.6 ($\pm$ 1.98) | 0.14 ($\pm$ 0.02) |
| $k$-means Embedding (Sener & Savarese, 2018) | 80.6 ($\pm$ 0.54) | 0.13 ($\pm$ 0.01) | 55.5 ($\pm$ 0.70) | 0.06 ($\pm$ 0.01) | 55.4 ($\pm$ 1.46) | 0.11 ($\pm$ 0.02) |
| Grad Matching (Campbell & Broderick, 2019) | 78.5 ($\pm$ 0.86) | 0.15 ($\pm$ 0.01) | 60.0 ($\pm$ 1.24) | 0.04 ($\pm$ 0.01) | 57.8 ($\pm$ 1.35) | 0.08 ($\pm$ 0.02) |
| GSS (Aljundi et al., 2019b) | 76.0 ($\pm$ 0.58) | 0.19 ($\pm$ 0.01) | 59.7 ($\pm$ 1.22) | 0.04 ($\pm$ 0.01) | 60.2 ($\pm$ 1.00) | 0.07 ($\pm$ 0.01) |
| ER-MIR (Aljundi et al., 2019a) | 80.7 ($\pm$ 0.72) | 0.14 ($\pm$ 0.01) | 60.2 ($\pm$ 0.72) | 0.04 ($\pm$ 0.00) | 56.9 ($\pm$ 2,25) | 0.11 ($\pm$ 0.03) |
| Bilevel Optim (Borsos et al., 2020) | 80.7 ($\pm$ 0.44) | 0.14 ($\pm$ 0.00) | 60.1 ($\pm$ 1.07) | 0.04 ($\pm$ 0.01) | 58.1 ($\pm$ 2.26) | 0.08 ($\pm$ 0.02) |
| OCS (Ours) | **82.5** ($\pm$ **0.32**) | **0.08** ($\pm$ **0.00**) | **60.5** ($\pm$ **0.55**) | **0.04** ($\pm$ **0.01**) | **61.5** ($\pm$ **1.34**) | **0.03** ($\pm$ **0.01**) |
| Multitask | 89.8 ($\pm$ 0.37) | – | 71.0 ($\pm$ 0.21) | – | 57.4 ($\pm$ 0.84) | – |
| **Imbalanced CL** | | | | | | |
| Finetune | 39.8 ($\pm$ 1.06) | 0.54 ($\pm$ 0.01) | 45.3 ($\pm$ 1.38) | 0.17 ($\pm$ 0.01) | 27.6 ($\pm$ 3.66) | 0.22 ($\pm$ 0.04) |
| Stable SGD (Mirzadeh et al., 2020) | 52.0 ($\pm$ 0.25) | 0.19 ($\pm$ 0.00) | 48.7 ($\pm$ 0.64) | 0.03 ($\pm$ 0.00) | 29.5 ($\pm$ 4.09) | 0.20 ($\pm$ 0.02) |
| Uniform Sampling | 61.6 ($\pm$ 1.72) | 0.15 ($\pm$ 0.01) | 51.0 ($\pm$ 0.78) | 0.03 ($\pm$ 0.00) | 35.0 ($\pm$ 3.03) | 0.11 ($\pm$ 0.03) |
| iCaRL (Rebuffi et al., 2017) | 71.7 ($\pm$ 0.69) | 0.09 ($\pm$ 0.00) | 51.2 ($\pm$ 1.09) | 0.02 ($\pm$ 0.00) | 43.6 ($\pm$ 2.95) | 0.05 ($\pm$ 0.03) |
| $k$-means Features (Nguyen et al., 2018) | 52.3 ($\pm$ 1.48) | 0.24 ($\pm$ 0.01) | 50.6 ($\pm$ 1.52) | 0.04 ($\pm$ 0.01) | 36.1 ($\pm$ 1.75) | 0.09 ($\pm$ 0.02) |
| $k$-means Embedding (Sener & Savarese, 2018) | 63.2 ($\pm$ 0.90) | 0.13 ($\pm$ 0.01) | 50.4 ($\pm$ 1.39) | 0.03 ($\pm$ 0.01) | 35.6 ($\pm$ 1.35) | 0.11 ($\pm$ 0.02) |
| Grad Matching (Campbell & Broderick, 2019) | 55.6 ($\pm$ 1.86) | 0.18 ($\pm$ 0.02) | 51.1 ($\pm$ 1.14) | 0.02 ($\pm$ 0.00) | 34.6 ($\pm$ 0.50) | 0.12 ($\pm$ 0.01) |
| GSS (Aljundi et al., 2019b) | 68.7 ($\pm$ 0.98) | 0.18 ($\pm$ 0.01) | 44.5 ($\pm$ 1.35) | 0.04 ($\pm$ 0.01) | 32.9 ($\pm$ 0.90) | 0.13 ($\pm$ 0.01) |
| ER-MIR (Aljundi et al., 2019a) | 69.3 ($\pm$ 1.01) | 0.16 ($\pm$ 0.01) | 44.8 ($\pm$ 1.42) | 0.03 ($\pm$ 0.01) | 32.3 ($\pm$ 3.49) | 0.15 ($\pm$ 0.03) |
| Bilevel Optim (Borsos et al., 2020) | 63.2 ($\pm$ 1.04) | 0.22 ($\pm$ 0.01) | 44.0 ($\pm$ 0.86) | 0.03 ($\pm$ 0.01) | 35.1 ($\pm$ 2.78) | 0.12 ($\pm$ 0.02) |
| OCS (Ours) | **76.5** ($\pm$ **0.84**) | **0.08** ($\pm$ **0.01**) | **51.4** ($\pm$ **1.11**) | **0.02** ($\pm$ **0.00**) | **47.5** ($\pm$ **1.66**) | **0.03** ($\pm$ **0.02**) |
| Multitask | 81.0 ($\pm$ 0.95) | – | 48.2 ($\pm$ 0.72) | – | 41.4 ($\pm$ 0.97) | – |

Figure 4: **(a)** Average accuracy **(b)** First task accuracy for balanced/imbalanced Rotated MNIST during CL.

## 5.2 QUANTITATIVE ANALYSIS FOR CONTINUAL LEARNING

**Balanced continual learning.** Table 1 shows the results on the balanced CL benchmarks. First, observe that compared to the random replay based methods (A-GEM and ER-Reservoir), OCS shows 19% relative gain in average accuracy, 62% and 79% reduction in forgetting over the strongest baseline on Rotated MNIST and Split CIFAR-100, respectively. Second, OCS reduces the forgetting by 38% and 57% on Rotated MNIST and Multiple Datasets respectively over the coreset-based techniques, demonstrating that it selects valuable samples from the previous tasks. We further illustrate this in Figure 4, where OCS consistently exhibits superior average accuracy and first task accuracy. Third, we show the scalability of OCS with larger episodic memory in Figure 5. Interestingly, iCaRL shows lower performance than uniform sampling with a larger memory buffer for Rotated MNIST, while OCS outperforms across all memory sizes on both datasets. Furthermore, we note that ER-MIR, GSS, and Bilevel Optim require $0.9\times$, $3.9\times$, and $4.2\times$ training time than OCS (see Table 5) on TITAN Xp, showing a clear advantage of OCS for the online streaming scenarios.

**Imbalanced continual learning.** To demonstrate the effectiveness of OCS in challenging scenarios, we evaluate on imbalanced CL in Table 1. We emphasize that compared to balanced CL, OCS shows significant gains over all the baselines for Rotated MNIST and Multiple Datasets. Notably, it leads to a relative improvement of $\sim 7\%$ and $\sim 9\%$ on the accuracy, $\sim 11\%$ and 40% reduction on the forgetting compared to the best baseline for each dataset, respectively. The poor performance of the baselines in this setup is largely attributed to their lack of current task coreset selection, which results in a biased estimate degenerating model performance (see Figure 8). Moreover, we observe that OCS outperforms Multitask for complex imbalanced datasets, perhaps due to the bias from the

Table 2: Performance comparison of OCS and other baselines on varying proportions of noise instances during noisy continual learning. We report the mean and standard-deviation of the average accuracy (Accuracy) and average forgetting (Forgetting) across five independent runs. The best results are highlighted in **bold**.

| Method | 0% | | 40% | | 60% | |
|---|---|---|---|---|---|---|
| | Accuracy | Forgetting | Accuracy | Forgetting | Accuracy | Forgetting |
| Stable SGD (Mirzadeh et al., 2020) | 70.8 ($\pm$ 0.78) | 0.10 ($\pm$ 0.02) | 56.2 ($\pm$ 0.95) | 0.40 ($\pm$ 0.01) | 56.1 ($\pm$ 0.62) | 0.40 ($\pm$ 0.01) |
| Uniform sampling | 79.9 ($\pm$ 1.32) | 0.14 ($\pm$ 0.01) | 74.9 ($\pm$ 2.45) | 0.20 ($\pm$ 0.03) | 68.3 ($\pm$ 3.68) | 0.26 ($\pm$ 0.03) |
| iCaRL (Rebuffi et al., 2017) | 80.7 ($\pm$ 0.44) | 0.13 ($\pm$ 0.00) | 77.4 ($\pm$ 0.60) | 0.18 ($\pm$ 0.01) | 71.4 ($\pm$ 2.63) | 0.23 ($\pm$ 0.03) |
| k-means embedding (Sener & Savarese, 2018) | 80.6 ($\pm$ 0.54) | 0.13 ($\pm$ 0.01) | 78.5 ($\pm$ 0.86) | 0.17 ($\pm$ 0.00) | 77.5 ($\pm$ 1.67) | 0.26 ($\pm$ 0.03) |
| GSS (Aljundi et al., 2019b) | 76.0 ($\pm$ 0.58) | 0.19 ($\pm$ 0.01) | 71.7 ($\pm$ 0.95) | 0.19 ($\pm$ 0.01) | 68.8 ($\pm$ 1.02) | 0.17 ($\pm$ 0.02) |
| ER-MIR (Aljundi et al., 2019a) | 80.7 ($\pm$ 0.72) | 0.14 ($\pm$ 0.01) | 76.0 ($\pm$ 1.34) | 0.17 ($\pm$ 0.01) | 73.5 ($\pm$ 0.94) | 0.18 ($\pm$ 0.01) |
| OCS (Ours) | **82.5** ($\pm$ 0.32) | **0.08** ($\pm$ 0.00) | **80.4** ($\pm$ 0.20) | **0.14** ($\pm$ 0.00) | **80.3** ($\pm$ 0.75) | **0.10** ($\pm$ 0.01) |

(a) Rotated MNIST      (b) Multiple Datasets

Figure 5: Performance comparison on various coreset sizes for balanced/imbalanced continual learning.

dominant classes and the absence of selection criteria in Multitask. Similar to balanced CL, OCS leads to superior performance for larger episodic memory in imbalanced CL (see Figure 5).

**Noisy continual learning.** Next, we evaluate on noisy Rotated MNIST dataset, which is constructed by perturbing a proportion of instances of the original dataset with Gaussian noise $\mathcal{N}(0, 1)$. Table 2 shows that the addition of noise significantly degrades the performance on all the baselines. In contrast, OCS leads to a relative gain of 43% on accuracy, 20% and 35% reduction in forgetting on 40% and 60% proportion of noisy data. Note that the performance gap is more significant for the higher distribution of noisy examples, supporting our claim that the similarity and diversity across the training examples in the coreset play an essential role for the task adaptation in continual learning.

## 5.3 ABLATION STUDIES

**Effect of gradients.** In Table 3, we empirically justify the utilization of gradients (Grad-OCS) compared to the raw inputs (Input-OCS) and feature-representations (Feat-OCS). We observe that Grad-OCS significantly outperforms Input-OCS and Feat-OCS on balanced and imbalanced CL, demonstrating that the gradients are a better metric to approximate the dataset.

**Effect of individual components.** We further dissect Minibatch similarity ($\mathcal{S}$), Sample diversity ($\mathcal{V}$) and Coreset affinity ($\mathcal{A}$) in Table 4. Note that selection using $\mathcal{S}$ shows reasonable performance as the model can select valuable data points; however, it may select redundant samples, which degrades its performance. In addition, $\mathcal{V}$ in isolation is insufficient since it can select non-redundant and non-representative instances. The combination of $\mathcal{S}$ and $\mathcal{V}$ improves the average accuracy, but it shows a marginal improvement on the forgetting. To further gain insight into $\mathcal{S}$ and $\mathcal{V}$, we interpolate between $\mathcal{S}$ and $\mathcal{V}$ in Figure 6, where we can observe that an optimal balance of $\mathcal{S}$ and $\mathcal{V}$ (indicated by the arrows) can further improve the performance of our proposed selection strategy.

Furthermore, $\mathcal{A}$ improves the forgetting since the selected candidates have similar gradient direction to the coreset of the previous tasks maximizing their performance. However, $\mathcal{A}$ does not consider the current task distribution explicitly and depends on the quality of the memorized replay buffer. We observe that using $\mathcal{A}$ in isolation obtains reasonably high performance on simple digit-based domain-incremental CL problems (e.g., Rotated MNIST) due to its inherent high resemblance among same class instances. Consequently, suppressing catastrophic forgetting by selective training based on $\mathcal{A}$ shows a relatively high impact rather than selective training based on distinguishing more informative or diverse samples. In contrast, $\mathcal{A}$ in isolation for selection is insufficient for complicated and realistic CL problems such as imbalanced CL and multiple datasets, and all the three components ($\mathcal{S}$, $\mathcal{V}$, and $\mathcal{A}$) contribute to the performance of OCS. For Multiple Datasets, selection using only $\mathcal{A}$ (58.1%) obtained worse performance in comparison to $\mathcal{S} + \mathcal{V} + \mathcal{A}$ (61.5%) and $\mathcal{S} + \mathcal{V}$ (58.6%). Further, selection using $\mathcal{S} + \mathcal{A}$ (59.4 $\pm$ 2.0%) and $\mathcal{V} + \mathcal{A}$ (56.4% $\pm$ 1.3) also obtained 2.1%p and 5.1%p lower average accuracy than full OCS, respectively.

Table 3: Ablation study for analyzing the effect of gradients selection for OCS.

| Method | Balanced Rotated MNIST | | Imbalanced Rotated MNIST | |
|---|---|---|---|---|
| | Accuracy | Forgetting | Accuracy | Forgetting |
| Input-OCS | 72.7 (± 0.47) | 0.13 (± 0.01) | 50.6 (± 1.74) | 0.04 (± 0.00) |
| Feat-OCS | 71.7 (± 0.62) | 0.17 (± 0.01) | 30.6 (± 0.40) | 0.03 (± 0.01) |
| Grad-OCS | **82.5** (± 0.32) | **0.08** (± 0.00) | **76.5** (± 0.84) | **0.08** (± 0.01) |

Table 4: Ablation study to investigate the impact of selection criteria $\mathcal{S}$, $\mathcal{V}$, and $\mathcal{A}$ on OCS.

| Method | | | Noisy Rot-MNIST (60%) | | Multiple Datasets | |
|---|---|---|---|---|---|---|
| $\mathcal{S}$ | $\mathcal{V}$ | $\mathcal{A}$ | Accuracy | Forgetting | Accuracy | Forgetting |
| ✓ | − | − | 64.0 (± 1.18) | 0.33 (± 0.01) | 56.3 (± 0.97) | 0.10 (± 0.02) |
| − | ✓ | − | 40.1 (± 1.32) | 0.06 (± 0.02) | 49.8 (± 1.30) | 0.12 (± 0.01) |
| − | − | ✓ | 79.6 (± 0.87) | 0.10 (± 0.01) | 58.1 (± 0.96) | 0.05 (± 0.01) |
| ✓ | ✓ | − | 66.8 (± 1.39) | 0.30 (± 0.01) | 58.6 (± 1.91) | 0.09 (± 0.02) |
| ✓ | ✓ | ✓ | **80.3** (± 0.75) | **0.10** (± 0.01) | **61.5** (± 1.34) | **0.03** (± 0.01) |

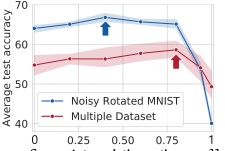

Figure 6: Interpolation between $\mathcal{S}$ and $\mathcal{V}$.

Table 5: Running time on Balanced Rot-MNIST.

| Method | Training Time |
|---|---|
| ER-MIR | 0.38 h (×0.87) |
| GSS | 1.71 h (×3.89) |
| Bilevel | 1.83 h (×4.17) |
| OCS (Ours) | 0.44 h (×1.00) |

Table 6: Collaborative learning with rehearsal-based CL on various datasets with 20 tasks each.

| | MC-SGD (Mirzadeh et al., 2021) | | MC-SGD + OCS | |
|---|---|---|---|---|
| Dataset | Accuracy | Forgetting | Accuracy | Forgetting |
| Per-MNIST | 84.6 (± 0.54) | 0.06 (± 0.01) | **86.6** (± 0.42) | **0.02** (± 0.00) |
| Rot-MNIST | 82.3 (± 0.68) | 0.08 (± 0.01) | **85.1** (± 0.27) | **0.04** (± 0.00) |
| Split CIFAR | 58.4 (± 0.95) | 0.02 (± 0.00) | **59.1** (± 0.55) | **0.00** (± 0.00) |

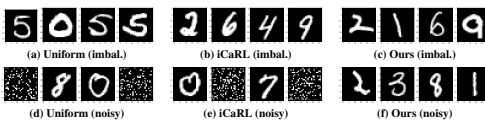

(a) Uniform (imbal.)   (b) iCaRL (imbal.)   (c) Ours (imbal.)

(d) Uniform (noisy)   (e) iCaRL (noisy)   (f) Ours (noisy)

Figure 7: Randomly picked coreset examples. Top: Imbalanced Rotated MNIST. Bottom: Noisy Rotated MNIST with 60% of noisy instances.

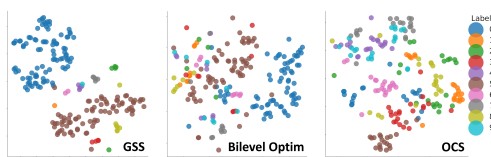

GSS   Bilevel Optim   OCS

Figure 8: T-SNE visualization of the selected samples on Imbalanced Rotated MNIST.

## 5.4 FURTHER ANALYSIS

**Coreset visualization.** Next, we visualize the coreset selected by different methods for imbalanced and noisy rotated MNIST in Figure 7. We observe that uniform sampling selects highly biased samples representing the dominant classes for imbalanced CL and noisy instances for noisy CL. In contrast, iCaRL selects the representative samples per class for imbalanced CL; however, it selects noisy instances during noisy CL. In comparison, OCS selects the beneficial examples for each class during imbalanced CL and discards uninformative noisy instances in the noisy CL training regime.

**T-SNE visualization.** We further compare the T-SNE visualization of the selected coreset by Bilevel Optim, GSS and OCS in Figure 8. We observe that the samples chosen by OCS are diverse, whereas Bilevel Optim and GSS select the majority of the samples from the dominant classes. We attribute the representative clusters and diversity in the samples selected by OCS to our proposed $\mathcal{S}$ (selects the valuable samples) and $\mathcal{V}$ (minimizes the redundancy among the selected samples) criteria.

**Collaborative learning with MC-SGD.** We remark that OCS can be applied to any rehearsal-based CL method with a replay buffer during training. We empirically demonstrate the effect of collaborative learning with other CL methods in Table 6. In particular, we use Mode Connectivity SGD (MC-SGD) (Mirzadeh et al., 2021), which encourages the mode connectivity between model parameters for continual and multitask loss and approximates the multitask loss through randomly selected replay buffer. Note that OCS leads to a relative gain of 1.2% to 3.4% on accuracy over MC-SGD on Permuted MNIST, Rotated MNIST, and Split CIFAR-100 datasets. Furthermore, MC-SGD + OCS shows considerably lower forgetting, illustrating that OCS prevents the loss of prior task knowledge.

## 6 CONCLUSION

We propose Online Coreset Selection (OCS), a novel approach for coreset selection during online continual learning. Our approach is modelled as a gradient-based selection strategy that selects representative and diverse instances, which are useful for preserving the knowledge of the previous tasks at each iteration. This paper takes the first step to utilize the coreset for improving the current task adaptation, while mitigating the catastrophic forgetting on previous tasks. Our experimental evaluation on the standard balanced continual learning datasets against state-of-the-art rehearsal-based techniques demonstrates the efficiency of our approach. We also show promising results on various realistic and challenging imbalanced and noisy continual learning datasets. We further show the natural extension of our selection strategy to existing rehearsal-based continual learning using a random-replay buffer. Our future work will focus on improving the selection strategies and exploring ways to utilize unlabelled data stream during training.

ACKNOWLEDGEMENT

This work was supported by the Engineering Research Center Program through the National Research Foundation of Korea (NRF) funded by the Korean Government MSIT (NRF-2018R1A5A1059921) and Institute of Information & communications Technology Planning & Evaluation (IITP) grant funded by the Korea government(MSIT) (No.2019-0-00075, Artificial Intelligence Graduate School Program(KAIST)).

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

**Organization.** The appendix is organized as follows: We first provide the experimental setups, including the dataset construction for balanced, imbalanced, noisy continual learning, and the hyperparameter configurations for OCS and all baselines in Appendix A. Next, we evaluate the selection criteria of the baselines for current task training and provide an additional ablation study comparing OCS with current task training to uniform selection in Appendix B.

## A  EXPERIMENTAL DETAILS

**Datasets.** We evaluate the performance of OCS on the following benchmarks:

1. **Balanced and Imbalanced Rotated MNIST.** These datasets are MNIST handwritten digits dataset (LeCun et al., 1998) variants containing 20 tasks, where each task applies a fixed random image rotation (between 0 and 180 degrees) to the original dataset. The imbalanced setting contains a different number of training examples for each class in a task, where we randomly select 8 classes over 10 at each task and each class contains $10\%$ of training instances for training. The total amount of training instances at each class can be [5900, **670**, **590**, **610**, **580**, 5400, **590**, **620**, **580**, **590**], where bold fonts denote the reduced number of instances for selected classes. The size of the replay buffer is 200 for all the rehearsal-based methods.

2. **Balanced and Imbalanced Split CIFAR-100.** These datasets are CIFAR-100 dataset (Krizhevsky, 2012) variants, where each task consists of five random classes out of the 100 classes. We use the Long-Tailed CIFAR-100 (Cui et al., 2019) for Imbalanced Split CIFAR-100 consisting of $n = n_i \mu^i$ samples for each class, where $i$ is the class index, $n_i$ is the original number of training images, and $\mu = 0.05$. It contains 20 tasks of five random classes out of the 100 classes. The size of the replay buffer is 100 (one example per class) for all the rehearsal-based methods.

3. **Balanced and Imbalanced Multiple Datasets.** This dataset contains a sequence of five benchmark datasets: MNIST (LeCun et al., 1998), fashion-MNIST (Xiao et al., 2017), NotM-NIST (Bulatov, 2011), Traffic Sign (Stallkamp et al., 2011), and SVHN (Netzer et al., 2011), where each task contains randomly selected 1000 training instances from each dataset. This dataset contains five tasks and 83 classes. We use the same strategy as Long-Tailed CIFAR-100 to construct the imbalanced Multiple Datasets with $\mu = 0.1$. The size of the replay buffer is 83 (one example per class) for all rehearsal-based methods.

**Network Architectures.** We use a MLP with 256 ReLU units in each layer for Rotated MNIST and a ResNet-18 (He et al., 2016) for Split CIFAR-100 datasets following Mirzadeh et al. (2020). For Rotated MNIST experiments, we use a single-head architecture, where the final classifier layer is shared across all the tasks, and the task identity is not provided during inference. In contrast, we use the multi-head structured ResNet-18 for CIFAR-100 and Multiple Datasets experiments, where the task identifiers are provided, and each task consists of its individual linear classifier.

**Implementations.** We follow the design of Mirzadeh et al. (2021) for evaluating all the methods. We utilize their implementation for Finetune, EWC (Kirkpatrick et al., 2017), Stable SGD (Mirzadeh et al., 2020), A-GEM (Chaudhry et al., 2019a), ER-Reservoir (Chaudhry et al., 2019b) and MC-SGD (Mirzadeh et al., 2021). We adapt the implementation released by Borsos et al. (2020) for Uniform Sampling, iCaRL (Rebuffi et al., 2017), $k$-means Features (Nguyen et al., 2018), $k$-means Embedding (Sener & Savarese, 2018), Grad Matching (Campbell & Broderick, 2019), and Bilevel Optim (Borsos et al., 2020). Further, we implement GSS (Aljundi et al., 2019b) and ER-MIR (Aljundi et al., 2019a) following the official code relased by the authors. Following iCaRL (Rebuffi et al., 2017), we store a balanced coreset $\mathcal{C}_t$ (equal number of examples per class) among the collected coreset candidates.

**Memory Capacity.** Existing rehearsal-based continual learning methods (Chaudhry et al., 2019a;b; Aljundi et al., 2019a; Mirzadeh et al., 2020; 2021) adopt two typical strategies to store data points in the replay buffer at the end of training each task: (1) memorizing a fixed number of data points per task ($|C_i| = J/T$), where an index of the task in a task sequence $i \in \{1, ..., T\}$, the total number of task $T$, and memory capacity $J$, (2) fully utilizing the memory capacity and randomly discard stored samples of each task when new data points arrive from next tasks ($|C_i| = J/t$), where the current task $t$ and an index of the observed task $i \in \{1, ..., t\}$. While both strategies are available, we adopt the latter strategy for all baselines and OCS. The details of this are omitted in Algorithm 1 for simplicity.

Table B.8: Performance comparison of baselines for current task adaptation. We report the mean and standard-deviation of the average accuracy and average forgetting across five independent runs.

| Method | Balanced Rotated MNIST | | Imbalanced Rotated MNIST | |
|---|---|---|---|---|
| | Accuracy | Forgetting | Accuracy | Forgetting |
| Finetune | 46.3 (± 1.37) | 0.52 (± 0.01) | 39.8 (± 1.06) | 0.54 (± 0.01) |
| Stable SGD | 70.8 (± 0.78) | 0.10 (± 0.02) | 52.0 (± 0.25) | 0.19 (± 0.00) |
| Uniform | 78.9 (± 1.16) | 0.14 (± 0.00) | 63.5 (± 1.09) | 0.14 (± 0.02) |
| iCaRL | 70.9 (± 0.82) | 0.12 (± 0.01) | 70.0 (± 0.60) | 0.10 (± 0.01) |
| $k$-means Feat. | 77.9 (± 1.08) | 0.15 (± 0.01) | 66.6 (± 2.42) | 0.12 (± 0.02) |
| $k$-means Emb. | 78.1 (± 1.53) | 0.14 (± 0.01) | 67.2 (± 0.16) | 0.10 (± 0.01) |
| Grad Matching | 79.0 (± 1.11) | 0.15 (± 0.01) | 52.4 (± 1.34) | 0.19 (± 0.02) |
| OCS (Ours) | **82.5** (± 0.32) | **0.08** (± 0.00) | **76.5** (± 0.84) | **0.08** (± 0.01) |
| Multitask | 89.8 (± 0.37) | — | 81.0 (± 0.95) | — |

Table B.9: Performance comparison of Class-incremental CL on balanced and imbalanced Split CIFAR-100. We report the mean and standard-deviation of the average accuracy and average forgetting across five independent runs.

| Method | Balanced Split CIFAR-100 | | Imbalanced Split CIFAR-100 | |
|---|---|---|---|---|
| | Accuracy | Forgetting | Accuracy | Forgetting |
| Finetune | 13.0 (± 0.38) | 0.33 (± 0.02) | 7.3 (± 0.31) | 0.15 (± 0.01) |
| Uniform | 16.4 (± 0.32) | 0.25 (± 0.01) | 8.7 (± 0.41) | 0.14 (± 0.01) |
| iCaRL | 18.1 (± 0.55) | 0.23 (± 0.01) | 8.6 (± 0.37) | 0.16 (± 0.01) |
| $k$-means Feat. | 16.6 (± 0.62) | 0.23 (± 0.01) | 8.9 (± 0.12) | 0.12 (± 0.02) |
| Grad Matching | 18.2 (± 0.70) | 0.22 (± 0.01) | 8.7 (± 0.36) | 0.23 (± 0.02) |
| ER-MIR. | 17.6 (± 0.35) | 0.22 (± 0.01) | 7.0 (± 0.44) | 0.10 (± 0.01) |
| OCS (Ours) | **20.1** (± 0.73) | **0.08** (± 0.01) | **11.1** (± 0.59) | **0.07** (± 0.00) |
| Multitask | 71.0 (± 0.21) | — | 48.2 (± 0.72) | — |

Table B.10: Performance comparison between uniform training with OCS coreset and original OCS method.

(a) Balanced Continual Learning

| | Uniform + OCS | | OCS | |
|---|---|---|---|---|
| Dataset | Accuracy | Forgetting | Accuracy | Forgetting |
| Rot-MNIST | 80.4 (± 0.61) | 0.14 (± 0.01) | **82.5** (± 0.32) | **0.08** (± 0.00) |
| CIFAR | 60.0 (± 1.30) | 0.04 (± 0.00) | **60.5** (± 0.55) | **0.04** (± 0.01) |
| Mul. Datasets | 56.3 (± 1.42) | 0.08 (± 0.03) | **61.5** (± 1.34) | **0.03** (± 0.01) |

(b) Imbalanced Continual Learning

| | Uniform + OCS | | OCS | |
|---|---|---|---|---|
| Dataset | Accuracy | Forgetting | Accuracy | Forgetting |
| Rot-MNIST | 73.6 (± 2.31) | 0.11 (± 0.01) | **76.5** (± 0.84) | **0.08** (± 0.00) |
| CIFAR | 51.3 (± 1.31) | 0.03 (± 0.01) | **51.4** (± 1.11) | **0.02** (± 0.00) |
| Mul. Datasets | 41.4 (± 2.51) | 0.05 (± 0.03) | **47.5** (± 1.66) | **0.03** (± 0.02) |

**Hyperparameter configurations.** Table A.7 shows the initial learning rate, learning rate decay, and batch size for each dataset that are shared among all the methods. Further, we report the best results obtained for $\lambda \in \{0.01, 0.05, 0.1, 1, 10, 50, 100\}$ for all the experiments. For OCS, we use batch size as 100 for Rotated MNIST and 20 for Split CIFAR-100 and Multiple Dataset. The running time reported in Table 5 was measured on a

Table A.7: Shared Hyperparameter configurations among our method and baselines for three datasets.

| Parameter | Rotated MNIST | Split CIFAR-100 | Multiple Datasets |
|---|---|---|---|
| Initial LR | 0.005 | 0.15 | 0.1 |
| LR decay | [0.75, 0.8] | 0.875 | 0.85 |
| Batch size | 10 | 10 | 10 |

single NVIDIA TITAN Xp. Due to the significant computational cost incurred by the training of Bilevel Optim (Borsos et al., 2020) for online continual learning, we restrict the bilevel optimization procedure to construct the replay buffer at the end of each task training.

**Choice of hyperparameters for OCS.** Note that we use the same value of $\kappa = 10, \tau = 1k$ for all experiments and analyses, including balanced, imbalanced, and noisy CL scenarios for all datasets, which shows that a simple selection of the hyperparameters is enough to show impressive performance. We expect careful tuning would further enhance the performance of OCS.

# B ADDITIONAL EXPERIMENTS

**Current task adaptation with the baselines.** One of our main contributions is the selective online training that selects the important samples for current task training. Therefore, we investigate the application of the other baselines for current task training in Table B.8. It is worth noting that all the rehearsal-based baselines that utilize their coreset selection criteria for current task adaptation decrease the performance (1.0 - 9.8%$p \downarrow$), except Grad Matching (0.5%$p \uparrow$) on Balanced Rotated MNIST. Moreover, for Imbalanced Rotated MNIST, Uniform Sampling, $k$-means Features, and $k$-means Embedding increase the performance 1.9%$p$, 13.3%$p$, and 4.0%$p$ compared to Table 1 respectively. In contrast, iCaRL and Grad Matching criteria decrease the performance by 1.7%$p$ and 3.2%$p$ on Imbalanced Rotated MNIST, respectively. On the contrary, OCS improves the performance for both the balanced and imbalanced scenarios. In light of this, we can conclude that efficient coreset selection plays a crucial role in imbalanced and noisy continual learning; therefore, the future rehearsal-based continual learning methods should evaluate their method on realistic settings rather than the standard balanced continual learning benchmarks.

**Class-incremental continual learning.** We also evaluate OCS for balanced and imbalanced class-incremental CL (CIL) setups in Table B.9. We Split CIFAR-100 dataset to five tasks with memory size of 1K for all methods. While the problem is extremely hard to solve, we want to emphasize that OCS outperforms the strongest baseline by 10.61% and 63.6% on accuracy and forgetting respectively

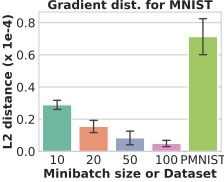 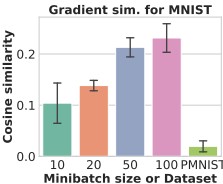 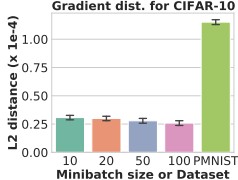 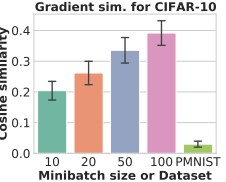

MNIST dataset using MLP                CIFAR-10 dataset using ResNet-18

**Figure B.9: Empirical validation of $\ell_2$ distance and cosine similarity between the gradient of the entire dataset and its minibatch gradient.** We report the mean and standard deviation of the metrics across five independent runs.

for balanced CIFAR-100 dataset, $24.05\%$ and $30\%$ on accuracy and forgetting respectively for imbalanced CIFAR-100 dataset in the CIL setting. Note that we report the results for all the baselines using hyperparameter configuration of the main experiments.

**Uniform training with OCS coreset.** We further analyze the effect of online coreset selection for current task training in Table B.10a. In particular, we compare uniform sampling for the current task while utilizing the coreset constructed by OCS for the previous tasks (Uniform + OCS) with our original selection scheme utilizing OCS for current and previous tasks. First, observe that Uniform + OCS shows $2.6\%$ and $9.2\%$ relative decrease in performance on Rotated MNIST and Multiple datasets respectively compared to our original OCS selection strategy. Second, note that Uniform + OCS significantly deteriorates the catastrophic forgetting for all datasets since uniformly sampled examples do not preserve the previous tasks knowledge. Moreover, imbalanced continual learning shows a similar trend in Table B.10b, where Uniform + OCS leads to a drop in both the accuracy and forgetting across all benchmarks. This further strengthens our claim that OCS is essential for the previous tasks and plays a vital role in encouraging current-task adaptation.

**OCS with partial gradients.** While we consider ResNet-18 as sufficiently deep neural networks, our OCS also can be utilized in extremely deep networks (e.g., ResNet-101) through OCS selection by computing partial gradients of the neural network to reduce the computational cost during online training. This simple modification is straightforward and to validate its potential, we have performed an additional ablation study on Split Tiny-

Table B.11: Ablation study for analyzing the selection with partial gradients in OCS.

| Used Blocks for OCS | Average Accuracy | Average Forgetting | Gradients Usage Ratio |
|---|---|---|---|
| [1,     ] | 35.72 ($\pm$ 0.57) | 5.45 ($\pm$ 0.55) | 1.6 |
| [1, 2,    ] | **37.42** ($\pm$ 0.93) | 4.70 ($\pm$ 0.94) | **6.3** |
| [1, 2, 3,   ] | 36.24 ($\pm$ 1.09) | 4.97 ($\pm$ 0.88) | **25.0** |
| [1, 2, 3, 4] | 37.06 ($\pm$ 0.69) | **4.32** ($\pm$ 0.71) | 100.0 |
| [   2, 3, 4] | 35.10 ($\pm$ 0.33) | 4.83 ($\pm$ 0.95) | 98.4 |
| [      3, 4] | 35.47 ($\pm$ 0.65) | 4.85 ($\pm$ 0.54) | 93.7 |
| [         4] | 34.75 ($\pm$ 1.16) | 5.20 ($\pm$ 0.43) | 75.0 |

ImageNet in Table B.11. ResNet-18 contains a convolutional layer with a first residual block (we name it as a block '1') and consecutive three other residual blocks (we name them as a block 2, 3, and 4, respectively). We have evaluated the variants of OCS which select the coreset based on weights gradients of partial blocks. The first column of the table describes gradients used in OCS selection. That is, the original OCS uses all gradients of all network components ([1,2,3,4]).

Surprisingly, we observe OCS using earlier two blocks (i.e., [1, 2]) show $0.36\%p$ higher performance while using only the $6.3\%$ of gradients compared to the original OCS. We expect that this specific benefit of earlier blocks is due to the different roles of blocks in neural networks. It is well known that earlier layers relatively focus on capturing generic representations while the latter ones capture class-discriminative information. Thus, obtaining the coreset that represents the task-general information and preserves shared knowledge with past tasks is specifically important since generic knowledge is easy to drift and much more susceptible to catastrophic forgetting.

We believe that further investigation of this observation would definitely provide more useful insights for future work and enhance the performance of OCS while greatly reducing the computational costs.

**Distance between the whole dataset and its minibatch.** To verify our conjecture that a minibatch can approximation of the whole dataset and select few representative data instances at each minibatch iteration, we empirically validate that the whole dataset and its minibatch have significant semantic relevancy. More formally, for a given subset $\mathcal{B}_t$, dataset $\mathcal{D}_t$, and $\epsilon > 0$, we suppose that the neural network satisfies following equation,

$$\mathcal{S}^*\left(\frac{1}{N_t}\nabla f_\Theta(\mathcal{D}_t), \frac{1}{|\mathcal{B}_t|}\nabla f_\Theta(\mathcal{B}_t)\right) \leq \epsilon, \qquad (10)$$

where $\mathcal{S}^*$ is an arbitrary distance function. To this end, we conduct a simple experiment using two different datasets (MNIST and CIFAR-10) and network architectures (MLP and ResNet-18) to show that it generally holds on various datasets and network architectures. We use a 2-layered MLP for MNIST and ResNet-18 for CIFAR-10 following our main experiments. At each iteration of training,we measure the $\ell_2$ distance ($\mathcal{S}^*(\cdot) = \ell_2(\cdot)$) and cosine similarity ($\mathcal{S}^*(\cdot) = 1/(\mathrm{sim}(\cdot))$) between the averaged gradient of the training minibatch and averaged gradient of the entire training dataset. To show that the distance is sufficiently small, we also measure the gradient $\ell_2$ distance between the entire dataset and the irrelevant dataset. Note that we recalculated the gradient for the whole dataset at each iteration for all results to correctly measure the distance at each step. As shown in Figure B.9, the gradient of larger minibatch shows better approximation to the gradient of the entire dataset for $\ell_2$ distance and cosine similarity. Further, note that that even the gradients of an arbitrary subset with a small-sized minibatch is significantly similar to the whole dataset with a small $\epsilon$, and it is more evident when compared to the irrelevant gradients from a different dataset, PMNIST (Permuted MNIST) (Goodfellow et al., 2013).

