# OpenReview forum: "Online Coreset Selection for Rehearsal-based Continual Learning"
_ICLR.cc/2022/Conference — ICLR 2022 Poster_

### Official Review · Reviewer_btdc · 2021-11-01

**Correctness:** 2
**Technical Novelty And Significance:** 3
**Empirical Novelty And Significance:** 3
**Recommendation:** 5
**Confidence:** 4

**Details Of Ethics Concerns:**

The paper does not involve ethical aspects.

**Main Review:**

Positive aspects:
- The proposed approach presents scientific novelty
- The related work section covers the most relevant papers in the field
- The experimental validation is extensive and the authors demonstrated the superiority of their approach.

Negative aspects:
- The idea is in general well-explained, although the paper lacks clarity in some aspects (see the detailed comments below), therefore it could be further improved
- There are also some issues with the proposed approach which are not clear enough

**Summary Of The Paper:**

The author propose a novel approach for online coreset selection, i.e exemplars used in the rehearsal process of past tasks in a continual learning framework. The proposed method is based on the observation that not all the samples in a dataset are equally valuable, but their quality affects model's effectiveness and efficiency. The method selects the most representative and informative samples at each iteration and trains them in an online manner. The approach has been conviently compared with state-of-the art methods and demonstrated its superiority.

**Summary Of The Review:**

Please find below my main concerns:
1. The following statement is not totally clear: "The naive CL design cannot retain the knowledge of previous tasks and thus results in catastrophic forgetting". What do you mean by 'naive CL design'? Please reformulate this statement.
2. Along the paper, you use repeatedly the expression 'target dataset'? What do you mean by 'target dataset' in a continual learning framework? It is confusing: If I have to learn 10 tasks, which one is the target dataset? I guess you refer to the 'current task'. Therefore, please use this formulation instead and change all over the document.
3. How many representative instances do you select from each mini batch?  Since a sample is presented several times during the training, shouldn't you use an acumulative measure and the final ranking/selection should be done at the end of the training? How do you guarantee the class-balance of the selected core-set? Please plot the distribution of samples per class resulted after the coreset selection process.
4. I did not understand the equation 3? What index 't' refers to: task ID? How is possible to measure the similarity between a sample and the minibatch it belongs to? Or you consider the similarity between a samples and the batch average? Something is missing there.
5. Equation 4: Confusing in terms of notations and terminology! You refer as 'cross-batch', but the eq. 4 is about the similarity between samples in the same batch! 'Cross-batch' should not refer to the similarity between a sample from one batch and the samples from different batches? Why the similarity measure in eq. 4 is negative? Please reconsider the notation and formulation of eqs 3 and 4.
6. Usually, in exemplar-based CL, the memory allocated for exemplar is known and fixed from the beginning. While new tasks are being learned, the number of exemplars from previous tasks is decreased in order to keep the memory size. What strategy do you adopt in your approach regarding the memory capacity where the exemplars are stored?

---

> ### Author Response · Authors · 2021-11-15
> **(2) Response for Reviewer btdc**
>
> > **[5-1]** *Equation 4: **Confusing in terms of notations and terminology! You refer as 'cross-batch', but the eq. 4** is about the similarity between samples in the same batch!*
>
> $\rightarrow$ By ‘cross-batch’, we refer to **the cross-sample diversity in a mini-batch** (a.k.a., cross-batch). However, we agree with you that it could be confusing to some readers. Thus, we will modify the terminology to **“sample diversity” in the final revision**. Thank you for the suggestion.
>
> ---
>
> > **[5-2]** ***Why is the similarity measure in eq. 4 negative?***
>
> $\rightarrow$ We formulate **the diversity of each data point $b\in\mathcal{B}$** as an **averaged dissimilarity** (i.e., an average of negative similarities) between a data point itself and other samples in the same minibatch $\mathcal{B}$, and not as an average similarity. Thus, the measure of sample diversity in Equation (4) is negative and the range is $[-1, 0]$. We have clarified this in a paragraph after Equation (4) in the revision **(page 5, highlighted in blue)**.
>
> ---
>
> > **[6]** *While new tasks are being learned, the number of exemplars from previous tasks is decreased in order to keep the memory size. **What strategy do you adopt in your approach regarding the memory capacity** where the exemplars are stored?*
>
> $\rightarrow$ Existing rehearsal-based continual learning methods (Chaudhry et al., 2019a;b; Aljundi et al., 2019a; Mirzadeh et al., 2020; 2021) adopt two typical strategies to store data points in the replay buffer at the end of training each task: **(1) memorizing a fixed number of data points per task** ($|C_i|=J/T$), where an index of the task in a task sequence $i\in\{1,...,T\}$, the total number of task $T$, and memory capacity $J$, **(2) fully utilizing the memory capacity** and **randomly discard stored samples of each task** when new data points arrive from next tasks ($|C_i|=J/t$), where the current task $t$ and an index of the observed task $i\in\{1,...,t\}$.
>
> While both strategies are available, we adopt **the latter strategy for all baselines and OCS**. While the details of this are omitted in Algorithm 1 for simplicity, we have clarified how we manage the memory capacity in the **‘Memory capacity’ paragraph in Section A of Appendix in our revision (highlighted in blue)**.

---

> ### Author Response · Authors · 2021-11-15
> **(1) Response for Reviewer btdc**
>
> Thank you for your time and efforts in reviewing our paper, and your constructive comments. We appreciate that you find our work as scientifically novel, superior over existing works, and based on extensive experimental validation. We clarify your concerns regarding several notations (‘naive CL design’, ‘target dataset’, and ‘cross-batch diversity’), Equation (3) and (4), and missing details on building the replay buffer below:
>
> ---
>
> > **[1]** ***What do you mean by 'naive CL design'?** Please reformulate this statement.*
>
> $\rightarrow$ By Naive CL design, we refer to **“a simple sequential training on multiple tasks without any means for tackling catastrophic forgetting”**. The objective of this statement was to describe the inaccessibility of the data points from the previous tasks in the standard continual learning setting with streaming data, and we have revised the statement in our revision to further clarify this point in **a paragraph after the equation (1) in Section 3** (highlighted the update in blue).
>
> ---
>
> > **[2]** *You use repeatedly the expression 'target dataset'? I guess you refer to the 'current task'.*
>
> $\rightarrow$ That is correct. The term **‘target task/dataset’** is a commonly used terminology in transfer learning to refer to **the task/dataset** that **receives knowledge transfer from the source task**. We used this term since for continual learning, the current task is the target task as the model’s goal is utilizing the previously learned knowledge from past tasks to enhance the model’s prediction performance on the current task. **We have revised the term** to avoid confusion in our revision (highlighted the update in blue):
> - **Line 13** in Abstract (Page 1)
> - **Line 6** in the second paragraph in Section 1 (Page 2)
> - **Line 4 - 5** in the third paragraph in Section 1 (Page 2, **two times**)
> - **Line 18** in the first paragraph in Section 3 (Page 3)
> - **Before Equation (2)** in Section 3 (Page 4)
> - **After Equation (4)** in Section 4 (Page 4, **two times**)
> - **After Equation (5)** in Section 4 (Page 5)
>
> ---
>
> > **[3-1]** ***How many representative instances do you select from each mini-batch?** Since a sample is presented several times during the training, shouldn't you use an accumulative measure and the final ranking/selection should be done at the end of the training?*
>
> $\rightarrow$ We select **$10$ instances from each mini-batch for all experiments** (Please see Table A.7). As described in a paragraph after Equation (6) in Section 4.1; (**"After the completion of each task training, we choose a coreset $\mathcal{C}_t$ among the collected candidates,"**), our method **performs the final coreset selection after the per-task training is complete**, but this is missing from Algorithm 1. We thank you for pointing this out. We have included a line for the final coreset selection **in Line 10 on Algorithm 1** in our revision (highlighted the update in blue).
> Moreover, we did not observe a significant performance disparity between iterative update of $\mathcal{C}_t$ and final selection from the accumulated candidates. We believe this is because **OCS avoids redundant sample selection** by considering **cross-batch diversity** criteria, which prevents the selection of the same (or similar) instances during training.
>
> ---
>
> > **[3-2]** ***How do you guarantee the class-balance of the selected coreset?***
>
> $\rightarrow$ For the final selection of the replay buffer $\mathcal{C}_t$ at the end of training each task $t$, we use a class-balanced selection motivated by iCaRL (Rebuffi et al., 2017). That is, **we obtain an equal number of examples per class** based on **OCS selection (Line 10 in Algorithm 1)** among the collected coreset candidates (Line 9 in Algorithm 1). (Please see the ‘Implementation’ paragraph in Appendix A).
> That is, after measuring the OCS score among the collected coreset candidates, we **obtain top-$k$ highest-scored data points at each class**, where $k= \lfloor|\mathcal{C}_t|/nc_t \rfloor$ and $nc_t$ is the number of classes of task $t$. (Line 10 in Algorithm 1).
>
> ---
>
> > **[4]** *Equation (3): What index does 't' refer to? task ID? **How is possible to measure the similarity between a sample and the minibatch it belongs to? Or do you consider the similarity between a sample and the batch average? Something is missing there.***
>
> $\rightarrow$ As described in the first paragraph in Section 3, $t$ is an index of the current task. **You may have missed the notation of $\overline{\nabla}f_{\Theta}$ in Equation (3)**, which denotes that we use **an average gradient of instances in a minibatch**. That is, we measure the similarity between the gradient from a sample and the averaged gradient from samples in a minibatch.
>
> ---
>
> [we continue our response below]

---

> ### Author Response · Authors · 2021-11-21
> **A gentle reminder**
>
> Dear reviewer btdc,
>
> We sincerely appreciate your comments that you find our work as scientifically novel, superior to existing works, and based on extensive experimental validation. We have made every effort to faithfully address all your comments in the responses.
> **We now have less than two days** to have interactive discussions. We have faithfully addressed your concerns regarding the term ‘naive CL design’ and ‘target dataset’, details for a selection of minibatch coreset and replay buffer, misunderstanding for Equation (3), and notations on Equation (4). The following is a quick summary of the response.
> -----
> - We have clarified the term **‘naive CL design’** that is “a simple sequential training on multiple tasks without any means for tackling catastrophic forgetting”.
> - We have revised the term **‘target datasets’** to ‘current datasets’ to avoid confusion in our revision.
> - We have explained the details on coreset selection per minibatch that OCS selects 10 instances from each mini-batch for all experiments and **performs the final coreset selection** after the per-task training is complete.
> - For the final selection of the replay buffer of the current task, we measure the OCS score among the collected coreset candidates and then obtain **top-$k$ highest-scored data points at each class**.
> - We have corrected the misunderstanding of reviewers in Equation (3). We believe that **the reviewer may have missed** the notation of **$\overline{\nabla}f_{\Theta}$ in Equation (3)**, which denotes that we use an **average gradient of instances** in a minibatch.
> - We have explained the term ‘cross-batch diversity’ that we refer to the **cross-sample diversity in a mini-batch** (a.k.a., cross-batch) and the reason for the negative form in Equation (4).
> - We have clarified the strategy for the construction of a replay buffer. For all baselines and OCS, we **fully utilize the memory capacity** and randomly discard stored samples of each task **when new data points arrive** from the next tasks.
> ----
> We sincerely appreciate your insightful and constructive comments and thank you again for your time and efforts in reviewing our paper. Please let us know if you have any further questions.
>
> Best regards,
> Authors

---

> ### Author Response · Authors · 2021-11-27
> **Dear Reviewer btdc - A Gentle Reminder**
>
> Dear reviewer btdc,
>
> We sincerely appreciate your efforts in reviewing our paper, and your constructive comments. We have responded to your comments and faithfully reflected them in the revision.
>
> As you know, now we have only a few days to have interactive discussions. Could you please go over our responses and the revision since end of the final discussion phase is approaching? Please let us know there is anything else we need to clarify or provide.
>
> Best,
> Authors

---

> ### Author Response · Authors · 2021-11-29
> **Dear Reviewer btdc - The end of the discussion period ends today.**
>
> Dear reviewer btdc,
>
> **Today is the end of the discussion deadline.** Could you please go over our rebuttal and check the responses? We believe that we have addressed all your concerns and that including these discussions will further strengthen our paper. **We hope you reflect this in your final review and the score.** We thank you again for your time and efforts in reviewing our paper.
>
> Thank you,
> Authors.

---

> > ### Comment · Reviewer_btdc · 2021-11-29
> > **Thank you for your responses**
> >
> > Dear Authors,
> >
> > I appreciate all your effort to clarify my concerns. Therefore, I am considering raising my initial rating.
> >
> > Thank you!

---

> > > ### Author Response · Authors · 2021-11-29
> > > **Thank you for your response, It seems you forgot to raise your score.**
> > >
> > > Thank you for your response! We thank the reviewer for reading our response and raising the score, but **it seems you forgot to raise your score**. Could you check your initial review and reflect the score correctly?

---

> > > > ### Comment · Reviewer_btdc · 2021-11-29
> > > > **Thank you for your response**
> > > >
> > > > This information is not available for you yet.

---

> > > > > ### Author Response · Authors · 2021-12-01
> > > > > **Dear Reviewer btdc, Could you finalize your decision reflecting our responses as you mentioned earlier?**
> > > > >
> > > > > Dear Reviewer btdc,
> > > > >
> > > > > We sincerely appreciate your effort for constructive feedback. One thing to note is that **there is no additional phase for the reviewers' final decision**, and now ACs are writing meta-reviews and recommending the submissions.
> > > > >
> > > > > **Could you finalize your decision reflecting our responses as you mentioned earlier?**
> > > > >
> > > > > We again thank you for your time and great effort.
> > > > >
> > > > > Best regards,
> > > > > Authors

---

### Official Review · Reviewer_XVBE · 2021-11-02

**Correctness:** 3
**Technical Novelty And Significance:** 3
**Empirical Novelty And Significance:** 3
**Recommendation:** 8
**Confidence:** 3

**Main Review:**

This paper proposes an Online Coreset Selection method to select the most representative and informative coreset at each iteration and trains them. The proposed method maximizes the model’s adaptation to a target dataset while selecting high-affinity samples to past tasks, which directly inhibits catastrophic forgetting.
1. The motivation of this manuscript is not clear. The authors should clearly claim the challenging issues in previous methods.
2. The authors complement the theoretical explanation of the success of the proposed approach.
3. While the online coreset selection method adopted in the manuscript seems plausible, it is not exciting.



**Summary Of The Paper:**

This paper proposes an Online Coreset Selection method to select the most representative and informative coreset at each iteration and trains them. The proposed method maximizes the model’s adaptation to a target dataset while selecting high-affinity samples to past tasks, which directly inhibits catastrophic forgetting. Experiments on the benchmark datasets show competitive results compared with baselines.

**Summary Of The Review:**

In general the paper is well organized and clearly written. The technical details are easy to follow. Experiments on the benchmark datasets show promising results compared with baselines.

---

> ### Author Response · Authors · 2021-11-15
> **Response for Reviewer XVBE**
>
> Dear reviewer,
> Thank you for your review and your thoughtful comments. We appreciate your comments that you find our paper is well organized and clearly written. We clarify your concerns regarding the clarification of the manuscript below.
>
> ---
>
> > ***Comment: The motivation of this manuscript is not clear.** The authors should clearly claim the challenging issues in previous methods.*
>
>
> $\rightarrow$ We believe the submission highlights the motivation of our OCS and challenging issues. We further remark the motivation of our work below:
> - **Realistic continual learning settings** in real-world applications often include the **imbalanced** number of instances per class or **noisy instances**, but many **prior CL works are missing** to consider these challenging scenarios (Please see the second paragraph in Section 1). Also, we observed that **each data point has a different impact on current task adaptation and catastrophic forgetting** of past tasks during continual learning (Please see Figure 3). Thus, **selecting the helpful coreset** to train from all the arriving data streams **is critical** to make a continual learner successfully adapt to the current task and prevent catastrophic forgetting of past tasks.
> - Meanwhile, the existing rehearsal-based methods (Rebuffi et al., 2017; Aljundi et al., 2019a;b; Chaudhry et al., 2019a;b) focus on memorizing informative instances as a replay buffer. They **do not select the coreset before the current task adaptation** and update the model on **all the arriving data streams**, which makes them susceptible to real-world applications due to **overfitting to a few dominant classes or training harmful noisy instances**. In contrast, OCS selects the instances before updating the model using our proposed selection criteria, which makes it robust to current task training across various practical CL scenarios (It is mentioned in the first paragraph in Section 2: 4~9th lines on Page 3).

---

> ### Author Response · Authors · 2021-11-21
> **A gentle reminder**
>
> Dear reviewer XVBE,
>
> We sincerely appreciate your positive comments that you find our paper is well organized and clearly written. We have made every effort to faithfully address all your comments in the responses.
> **We now have less than two days** to have interactive discussions. We have faithfully addressed your concerns regarding the motivation of our method. The following is a quick summary of the response.
> -----
> - Realistic CL settings in real-world applications often **include the imbalanced number of instances per class or noisy instances**.
> - Also, we observed that each data point **has a different impact** on **current task adaptation** and **catastrophic forgetting** of past tasks during CL. To this end, selecting the helpful coreset to train from all the arriving data streams is critical.
> - Existing methods update the model on all the arriving data streams, which makes them susceptible to real-world applications due to overfitting to a few dominant classes or training harmful noisy instances.
> - To tackle these challenges, OCS selects only **helpful instances before updating the model** using our proposed selection criteria that selected instances are **expressive** (minibatch similarity), **non-redundant** (cross-batch diversity), and **not harmful to past tasks’ performance** (coreset affinity), and updates the model using only selected coresets.
> -----
> We sincerely appreciate your insightful and constructive comments and thank you again for your time and efforts in reviewing our paper. Please let us know if you have any further questions.
>
> Best regards,
> Authors

---

> ### Author Response · Authors · 2021-11-27
> **Dear Reviewer XVBE - A Gentle Reminder**
>
> Dear reviewer XVBE,
>
> We sincerely appreciate your efforts in reviewing our paper, and your constructive comments. We have responded to your comments and faithfully reflected them in the revision.
>
> As you know, now we have only a few days to have interactive discussions. Could you please go over our responses and the revision since end of the final discussion phase is approaching? Please let us know there is anything else we need to clarify or provide.
>
> Best,
> Authors

---

> ### Author Response · Authors · 2021-11-29
> **Dear Reviewer XVBE - The end of the discussion period ends today.**
>
> Dear reviewer XVBE,
>
> **Today is the end of the discussion deadline.** Could you please go over our rebuttal and check the responses? We believe that we have addressed all your concerns and that including these discussions will further strengthen our paper. **We hope you reflect this in your final review and the score.** We thank you again for your time and efforts in reviewing our paper.
>
> Thank you,
> Authors.

---

> ### Comment · Reviewer_XVBE · 2021-11-29
> **Thanks for the responses**
>
> I really appreciate the authors for performing extra experiments and providing extra details to resolve some of the concerns. Therefore, I think I'd like to improve my rating.
>
> Thanks

---

> > ### Author Response · Authors · 2021-11-29
> > **Thank you for the response**
> >
> > We want to thank the reviewer for reading our response and raising the score. We are happy to hear that our extra experiments and details resolved the raised concerns.

---

### Official Review · Reviewer_iADy · 2021-11-02

**Correctness:** 3
**Technical Novelty And Significance:** 3
**Empirical Novelty And Significance:** 3
**Recommendation:** 6
**Confidence:** 4

**Details Of Ethics Concerns:**

The reviewer has no concerns.

**Main Review:**

STRENGTHS:
1) The paper is very well written and presented.
2) The core-set problem is an important and under researched area in continual learning, especially in its online form.
3) Evaluation is very well executed.
4) The introduction of diversity as a criteria for core-set selection is interesting.

WEAKNESSES:
1) As the term “online” appears in the title of the paper, this modality should be better introduced and motivated. A clear explanation appears in the related work at the end of the continual learning paragraph. However, the term remains not properly defined and seems to be related with the problem of imbalance.
What exactly does imbalance mean? In the reviewer understanding the term seems to be defined as in [*] and in (Aljundi 2019b), that is, task distribution is not i.i.d. Given that definition, why selecting the core-set before should provide an advantage? The reviewer understands that the performance is improved, however, the writing states that selecting the core-set before has a dependency on the imbalanced task distribution and that this is an advantage with respect to (Rebuffi et al., 2017; Aljundi et al., 2019b;a; Chaudhry et al., 2019a;b). If this is not the case and the motivation of selecting the core-set before adaptation is because of the good empirical results, then it should be better remarked in the paper.

[*] Chrysakis, Aristotelis, and Marie-Francine Moens. "Online continual learning from imbalanced data." International Conference on Machine Learning. PMLR, 2020.

2) Figure 2 shows a dataset that is not addressed by the method and is somewhat misleading. The "Multidataset"  used in the paper does not include the CIFAR dataset (i.e. complex objects like dogs and vehicles).

3) In the reviewer's opinion the claim of large improvement, included in the third contribution, seems to be somewhat a bit bold. For example, in CIFAR-100 balanced and unbalanced learning settings in Tab.1, the performance does not differ too much from the herding strategy used in iCARL (i.e. 60.3 vs 60.5, 51.2 vs 51.4). Although the reviewer noticed that in rotated MNIST and Multidataset improvements are evident, these datasets do not typically “transfer” their performance to larger datasets (i.e., ImageNet) as CIFAR-100 typically does.

4) The validating hypothesis of Fig.3 (i.e. learning from MNIST to CIFAR10) seems to favor diversity, which is exactly what herding is not doing. In herding, examples are selected closer to the class mean of the feature representation in each class. Herding is somewhat orthogonal to diversity. This may partly explain the not improving performance on CIFAR-100. This part should be discussed in depth (i.e., motivate that the method does not mostly favor datasets with diversity).

5) In the reviewer’s opinion the ablation study of the effect of the gradient is not sufficient to justify its usage. As also remarked in the point 3) of this review, the MNIST family datasets typically do not “transfer” their performance to more complex and bigger datasets. In other words, using the gradient on MNIST does not imply that the gradient on CIFAR or bigger datasets is a good choice.
Is the gradient computationally expensive for deeper neural networks? For example, what happens with one or two more orders of parameters?


**Summary Of The Paper:**

The paper presents three gradient-based selection criteria to select the core-set for improving adaptation and reducing catastrophic forgetting. Differently from other methods, the proposed approach selects the instances before updating the model.

**Summary Of The Review:**

This is a nice and well written paper in which some details need to be clarified. Specifically, imbalance, selecting the core-set before/after adaptation, dataset diversity and selection, the "transferability" of performance of the datasets.

---

> ### Author Response · Authors · 2021-11-17
> **(3) Response for Reviewer iADy**
>
> > **[5-1]** ***The ablation study of the effect of the gradient is not sufficient to justify its usage**. As also remarked in the point 3) of this review, the MNIST family datasets typically do not “transfer” their performance to **more complex and bigger datasets**. In other words, **using the gradient on MNIST does not imply that the gradient on CIFAR or bigger datasets is a good choice**.*
>
> $\rightarrow$ We want to emphasize that the gradient-based coreset selection strategies (Aljundi et al., 2019 a;b; Campbell & Broderick, 2019, Borsos et al., 2020) have been utilized in prior works and have been strong baselines to construct replay buffer for rehearsal-based continual learning.
>
> As followed by the reviewer’s suggestion, we additionally provide the ablation study using **split-TinyImageNet** dataset in Table below, where the experimental setting is the same as in comment [3]. We consistently observe that **gradient-based OCS shows superior performance to raw image- and feature-based counterparts**.
>
> |               		| Accuracy   		| Forgetting		|
> |---------------		|------------------		|------------------		|
> | Input-OCS		| 34.98 $\pm$ 1.23	| 6.50 $\pm$ 0.57 	|
> | Feats-OCS     	| 33.39 $\pm$ 0.76 	| 6.40 $\pm$ 0.84 	|
> | **Grads-OCS (Ours)**	| **37.06 $\pm$ 0.69**	| **4.32 $\pm$ 0.71**	|
>
> ---
>
> > **[5-2]** ***Is the gradient computationally expensive** for deeper neural networks? For example, what happens with one or two more orders of parameters?*
>
>
> $\rightarrow$ As described in Table 5, our OCS shows reasonable training speed compared to other baselines. We note that ER-MIR, GSS, and Bilevel Optim require 0.9x,3.9x, and 4.2x training time than OCS on TITAN Xp, showing a clear advantage of OCS for the online streaming scenarios.
>
> While we consider ResNet-18 as sufficiently deep neural networks, our OCS also can be utilized in extremely deep networks (e.g. ResNet-101) through OCS selection by computing partial gradients of the neural network to reduce the computational cost during online training. This simple modification is straightforward and to validate its potential, we have performed an additional ablation study on Split TinyImageNet:
>
> | Grad Blocks	| Accuracy   		| Forgetting		| Grads usage ratio |
> |---------------	|------------------		|------------------		|------------------	|
> | [1$~~~~~~~~$]     	| 35.72 $\pm$ 0.57 	| 5.45 $\pm$ 0.55 	| $~~~$1.6 % 	|
> | **[1,2$~~~~~~$]**    	| **37.42 $\pm$ 0.93** 	| **4.70 $\pm$ 0.94** 	| $~~~$**6.3 %**		|
> | **[1,2,3$~~~$]**    	| **36.24 $\pm$ 1.09** 	| **4.97 $\pm$ 0.88** 	| $~~~$**25.0 %**	|
> | **[1,2,3,4]**	| **37.06 $\pm$ 0.69**	| **4.32 $\pm$ 0.71** 	| $~~~$**100.0 %**	|
> | [$~~~$2,3,4]    	| 35.10 $\pm$ 0.33 	| 4.83 $\pm$ 0.95 	| $~~~$98.4 %	|
> | [$~~~~~~$3,4]    	| 35.47 $\pm$ 0.65 	| 4.85 $\pm$ 0.54 	| $~~~$93.7 %	|
> | [$~~~~~~~~~$4]	| 34.75 $\pm$ 1.16 	| 5.20 $\pm$ 0.43 	| $~~~$75.0 %	|
>
> ResNet-18 contains a convolutional layer with a first residual block (we name it as a block ‘1’) and consecutive three other residual blocks (we name them as a block 2, 3, and 4, respectively). We have evaluated the variants of OCS which select the coreset based on weights gradients of partial blocks in the neural network. The first column of the table describes gradients used in OCS selection. That is, the original OCS uses all gradients of all network components ([1,2,3,4]).
>
> Surprisingly, we observe **OCS using earlier two blocks (i.e., [1, 2])  show $0.36$%p higher performance** while using only the **$6.3$% of gradients** compared to the original OCS. We expect that this specific benefit of earlier blocks is due to the different roles of blocks in neural networks. It is well known that **earlier layers relatively focus on capturing generic representations** while the latter ones capture class-discriminative information. Thus, obtaining the coreset that represents the **task-general information** and preserves **shared knowledge with past tasks** is specifically important since generic knowledge is easy to drift and much more susceptible to catastrophic forgetting.
>
> We believe that further investigation of this observation would definitely provide more useful insights for future work and enhance the performance of OCS while greatly reducing the computational costs for gradients.

---

> ### Author Response · Authors · 2021-11-17
> **(2) Response for Reviewer iADy**
>
> > **[3]** ***The minor increase in the performance for CIFAR-100.** Although the reviewer noticed that in rotated MNIST and Multidataset improvements are evident, these datasets do not typically “transfer” their performance to larger datasets (i.e., ImageNet) as CIFAR-100 typically does.*
>
> $\rightarrow$ Note that the **scenario of multiple datasets is essential in practice**, which **simulates** the knowledge transfer and preservation **on various different task distributions**; gray-digits/fonts (MNIST and NotMNIST), colored-digits (SVHN), gray-objects (FashionMNIST), and colored-objects (Traffic signs).
>
> Further, we provide the substantially higher performance of our OCS on **split CIFAR-100 for class-incremental learning (CIL) setup** (Please see Table B.9 and a paragraph ‘Class-incremental continual learning’ in Appendix B): “OCS outperforms the strongest baseline by $10.61$% and $63.6$%on accuracy and forgetting respectively for balanced CIFAR-100 dataset, $24.05$% and $30$% on accuracy and forgetting respectively for imbalanced CIFAR-100 dataset in the CIL setting”.
>
> Nevertheless, we **additionally compare** our method on **sequential TinyImagenet** to show its scalability to larger datasets during the rebuttal period. TinyImagenet contains 64×64 downsized and colored images of $200$ classes and we design sequential TinyImagenet containing a sequence of $10$ tasks, where each task consists of $20$ classes out of $200$ classes. Each class has $500$ training images, and $50$ test images. We use ResNet-18 and we perform single epoch task-incremental learning with the hyperparameter setup as follows: init_lr$=0.01,$ lr_decay$:1.0, \lambda\in{0.01, 0.1, 1, 10}, \tau=10$ for all experiments. We report the mean and standard deviation of the average accuracy and average forgetting across **five independent runs**.
>
> As shown in the table below, OCS obtains **$3.2$% higher average accuracy and $29.1$% lower average forgetting** in comparison with the best-performed baseline. To this end, based on exhaustive empirical evaluations for Split CIFAR-100 (class-incremental learning), Multiple Datasets, and Split TinyImagenet, we have demonstrated that **our OCS successfully deals with the transfer to the upstream/larger datasets**.
>
> |               	| Accuracy   		| Forgetting		|
> |---------------	|------------------		|------------------		|
> | Uniform	| 34.98 $\pm$ 0.76 	| 7.35 $\pm$ 0.71 	|
> | iCaRL 	| 35.63 $\pm$ 0.80 	| 6.18 $\pm$ 0.58 	|
> | ER-MIR        	| 35.91 $\pm$ 0.63 	| 6.09 $\pm$ 0.83 	|
> | **OCS (Ours)** 	| **37.06 $\pm$ 0.69** | **4.32 $\pm$ 0.71** 	|
>
> ---
>
>
> >**[4-1]** *The validating hypothesis of Fig.3 (i.e. learning from MNIST to CIFAR10) seems to **favor diversity, which is exactly what herding is not doing**.*
>
> $\rightarrow$ We clarify that Figure 3 shows that few data points are much more robust to catastrophic forgetting than others. Hence, we believe that Figure 3 is **unrelated to diversity** and **even proximity** to the mean of examples **across tasks** (a.k.a., herding, used in iCaRL). We would appreciate your further comments to elucidate the details of your conclusion and would do our best to resolve them in the discussion.
>
> ---
>
> > **[4-2]** ***Cross-batch diversity is orthogonal to herding.** This may partly explain the not improving performance on CIFAR-100.*
>
> $\rightarrow$ First of all, we want to emphasize that **herding is not the optimal strategy to select expressive sub-samples**, specifically when the data distribution is wide and imbalanced due to an imbalance of dataset or noisy instances. Further, **herding often shows similar performance to random selection** (Please see Figure 1 in https://arxiv.org/pdf/1807.02802.pdf).
> As described in our response to the comment **[1-2]**, **considering diversity is also essential for coreset selection**. Recent rehearsal-based continual learning methods (Aljundi et al., 2019 a;b; Borsos et al., 2020)  construct the replay buffer by considering different measures of diversity among data points. Also, we empirically demonstrate the positive impact of cross-batch diversity in Table 4 and Figure 6 in the paper.

---

> ### Author Response · Authors · 2021-11-17
> **(1) Response for Reviewer iADy**
>
> Dear reviewer,
> Thank you for your review and your thoughtful comments. We appreciate that you find our coreset problem crucial for online CL and believe the paper is a well-written submission. We clarify your concerns regarding the term ‘online’ and ‘imbalanced’, the applicability to larger datasets, the effect of cross-batch diversity, and the impact of gradient-based selection below.
>
> ---
>
> > **[1-1]** ***The term “online” should be better introduced and motivated.** A clear explanation appears in the related work at the end of the continual learning paragraph. However, the term remains not properly defined and seems to be related to the problem of imbalance.*
>
> $\rightarrow$ We want to note that the term ‘Online’ in our paper follows the online continual learning setting from prior works (Aljundi et al., 2019a;b; Mirzadeh et al., 2021). Specifically, the term ‘Online’ in the title implies that OCS selects the batch coreset from arriving streaming data in an online fashion (Line 4-6 at Algorithm 1) and is not directly related to the problem of imbalance. We clarified it in the last line on page 1 in Section 1 in our revision (update is highlighted in blue).
>
> ---
>
>
> > **[1-2]** ***What exactly does imbalance mean?** why selecting the core-set before should provide an advantage? The writing states that selecting the core-set before has a dependency on the imbalanced task distribution and that this is an advantage with respect to (Rebuffi et al., 2017; Aljundi et al., 2019a;b; Chaudhry et al., 2019a;b).*
>
>
> $\rightarrow$ As described in Figure 2 (a) (and its caption) and mentioned in your review, we follow the definition  ‘imbalance’  from existing works ([*] and Aljundi 2019b), where each task has a different number of instances per class (e.g., imbalanced Rotated MNIST, imbalanced CIFAR-100, and imbalanced Multiple Datasets) or have a different number of classes (imbalanced Multiple Datasets).
>
> We consider the imbalance setting to simulate realistic scenarios that are missing from many prior CL works. Further, selecting the coreset before the current task adaptation is crucial for real-world applications which often include the imbalanced number of instances per class or noisy instances. The existing rehearsal-based methods (Rebuffi et al., 2017; Aljundi et al., 2019a;b; Chaudhry et al., 2019a;b) focus on memorizing informative instances as a replay buffer. They **do not select the coreset before the current task adaptation** and update the model on **all the arriving data streams**, which makes them susceptible to real-world applications due to **overfitting to a few dominant classes or training harmful noisy instances**. In contrast, OCS selects the instances before updating the model using our proposed selection criteria, which makes it robust to current task training across various practical CL scenarios.
>
> ---
>
>
> > **[2]** ***Figure 2 shows a dataset that is not addressed by the method and is somewhat misleading.** The "Multidataset" used in the paper does not include the CIFAR dataset (i.e. complex objects like dogs and vehicles).*
>
> $\rightarrow$ We want to note that Figure 2 is **a concept figure** to illustrate scenarios for imbalanced and noisy continual learning and **does not represent a real dataset** that we used in our main experiments. If you don’t like that, we can revise the figure for the real dataset construction we used, but we do not believe the current version of concept figures is misleading to readers.

---

> ### Author Response · Authors · 2021-11-21
> **A gentle reminder**
>
> Dear reviewer iADy,
>
>
> We sincerely appreciate your positive comments that you find our coreset problem crucial for online CL and believe the paper is a well-written submission. We have made every effort to faithfully address all your comments in the responses.
> **We now have less than two days** to have interactive discussions. We have faithfully addressed your concerns regarding the term ‘online’ and ‘imbalanced’, the applicability to larger datasets, the effect of cross-batch diversity, and the impact of gradient-based selection. The following is a quick summary of the response.
> ----
> - We have clarified the term ‘online’ in the original paper that ‘Online’ in the title implies that OCS selects the batch coreset from arriving streaming data in an online fashion.
>
> - We have clarified the term ‘imbalanced’ in the original paper that each task has a different number of instances per class or has a different number of classes.
>
> - We have demonstrated that **our OCS successfully deals with the transfer to the upstream/larger datasets**. For the Sequential TinyImagenet dataset, our OCS obtains **$3.2$% higher average accuracy and $29.1$% lower average forgetting** in comparison with the best-performed baseline.
>
> - We have clarified the positive impact of cross-batch diversity and its necessity.
> - We have performed an additional ablation study using the Sequential TinyImagenet that **validates the effectiveness of a gradient-based coreset selection strategy on larger datasets**.
> - We have clarified that our OCS shows reasonable training time compared to recent baselines. Also, we have provided a simple yet promising modification for scalable online selection. Obtaining coresets based on **partial gradients** from only a few blocks in larger deep neural networks show **0.36%p higher performance** while using only the **$6.3$% of gradients** compared to the original OCS. We believe that this observation would enhance the performance of OCS while greatly reducing the computational costs for gradients.
>
> ------
>
> We sincerely appreciate your insightful and constructive comments and thank you again for your time and efforts in reviewing our paper. Please let us know if you have any further questions.
>
> Best regards,
> Authors

---

> ### Author Response · Authors · 2021-11-27
> **Dear Reviewer iADy - A Gentle Reminder**
>
> Dear reviewer iADy,
>
> We sincerely appreciate your efforts in reviewing our paper, and your constructive comments. We have responded to your comments, faithfully reflected them in the revision, and provided additional experimental results that you have requested.
>
> As you know, now we have only a few days to have interactive discussions. Could you please go over our responses and the revision since end of the final discussion phase is approaching? Please let us know there is anything else we need to clarify or provide.
>
>
> Best,
> Authors

---

> ### Author Response · Authors · 2021-11-29
> **Dear Reviewer iADy - The end of the discussion period ends today.**
>
> Dear reviewer iADy,
>
> **Today is the end of the discussion deadline.** Could you please go over our rebuttal and check the responses? **We believe that we have addressed all your concerns, providing all additional experimental results requested,** and that including these discussions and experimental results will further strengthen our paper. **We hope you reflect this in your final review and the score.** We thank you again for your time and efforts in reviewing our paper.
>
> Thank you,
> Authors.

---

> > ### Comment · Reviewer_iADy · 2021-11-29
> > **clarifications...**
> >
> > The reviewer appreciates the clarifications to the concerns raised and he is considering them for the final rating.

---

> > > ### Author Response · Authors · 2021-11-29
> > > **Thank you for your response**
> > >
> > > We want to thank the reviewer for reading our response with further clarifications and considering them for the final rating.
> > > Thank you, Authors

---

> > > ### Author Response · Authors · 2021-12-01
> > > **Dear Reviewer iADy - Could you finalize your decision reflecting our responses as you mentioned earlier?**
> > >
> > > Dear Reviewer iADy,
> > >
> > > We sincerely appreciate your effort for constructive feedback. One thing to note is that **there is no additional phase for the reviewers' final decision**, and now ACs are writing meta-reviews and recommending the submissions.
> > >
> > > **Could you finalize your decision reflecting our responses as you mentioned earlier?**
> > >
> > > We again thank you for your time and great effort.
> > >
> > > Best regards,
> > > Authors

---

### Official Review · Reviewer_mR25 · 2021-11-03

**Correctness:** 4
**Technical Novelty And Significance:** 3
**Empirical Novelty And Significance:** Not applicable
**Recommendation:** 6
**Confidence:** 4

**Main Review:**

The reviewer lists the major strengths and weaknesses as follows.

1. Strengths:
This paper is well-structured and contains sufficient experiments. Its motivation is meaningful and interesting.

2. Weaknesses:
a. The authors did not explain how big the difficulty is to adapt existing algorithms to online learning. And the way of designing the proposed algorithm for online learning seems straightforward and cannot be regarded as a genuine technical contribution.

b. The algorithm lacks novelty. The core part of the proposed algorithm seems very similar to the core part of 'Asymmetric Multi-task Learning Based on Task Relatedness and Loss'. The three adopted selection strategies do not demonstrate enough novelty, either.

c. Some grammatical errors and typos exist, such as 'let .. is' in definition 3.


**Summary Of The Paper:**

This paper addresses the problem of coreset selection for realistic and challenging continual learning scenarios. The authors proposed Online Coreset Selection (OCS), a simple yet effective online coreset selection method to obtain a representative and diverse subset.

**Summary Of The Review:**

In a nutshell, the reviewer regards this paper as a borderline paper, given the limited technical innovation.

I read the authors' rebuttal. Some of my questions and concerns are replied well. However, I think that the core technique of this work does not advance the research in continual learning significantly. So, I adhere to my previous rating.

---

> ### Author Response · Authors · 2021-11-17
> **Response for Reviewer mR25**
>
> Dear reviewer,
> Thank you for your review and your constructive comments. We appreciate your comments that our paper is well structured, contains sufficient experiments, and comes with meaningful and interesting motivation. We clarify your concerns regarding the difficulty of adapting existing algorithms to online learning and the novelty of the algorithm below.
>
> ---
>
> > **[1]** *Explanation on **how big the difficulty is to adapt existing algorithms to online learning**. And the way of designing the proposed algorithm for online learning seems straightforward and **cannot be regarded as a genuine technical contribution**.*
>
> $\rightarrow$ We described the difficulty of adapting existing algorithms to online learning in the first paragraph in Section 2: the existing rehearsal-based methods (Rebuffi et al., 2017; Aljundi et al., 2019a;b; Chaudhry et al., 2019a;b) **update the model on all the arriving data streams**. In this realistic online continual learning scenarios, these existing methods **suffer from sub-optimal convergence or biased training due to noisy and class-imbalanced data points** in the arriving data streams. In contrast, OCS selects the instances using our proposed selection criteria prior to updating the model, which makes it robust to such noise and bias in various practical CL scenarios.
>
> Moreover, to discern the advantage of our OCS in the online learning setting compared to baselines, we **did provide** the experimental results for the **online learning version of existing rehearsal-based methods** in **Table B.8** and the **first paragraph in Appendix Section B** in the original paper. Note that **utilizing their selection criteria** for sample selection from arriving current task data stream **in an online manner decreases the performance** of the baselines except for Grad Matching. This **highlights the clear contribution and effectiveness in our OCS** for online selection of helpful coresets during continual learning.
>
> ---
>
> > **[2]** ***The algorithm lacks novelty.** The core part seems very similar to 'Asymmetric Multi-task Learning Based on Task Relatedness and Loss'. The three adopted selection strategies do not demonstrate enough novelty, either.*
>
> $\rightarrow$ This is a misunderstanding, since our method is based on a completely different motivation and idea from AMTL [Lee et al. 16], except that we also consider imbalanced dataset scenario.
>
> AMTL tackles the **multi-task learning problem** and propose a method that **performs asymmetric knowledge transfer across tasks** depending on the **task loss**. Specifically, AMTL transfers the knowledge from the task-specific model with low loss tasks to ones with higher loss tasks to prevent negative transfer from unreliable predictors (with high loss). To tackle this problem, AMTL adopts a regularization term that **minimizes the L2 distance among task weights** with a learnable asymmetric matrix $B$ to control the degree of knowledge transfer:
>
> $\underset{W,B\geq 0}{minimize}\sum^{T}_{t=1}\{(1+\mu\|\|b^o_t\|\|_1)\mathcal{L}(w_t;D_t) +\lambda||w_t -\sum^T_\{s=1\} B_\{st\} w_s||^2_2}$    (Equation (1) in ATML paper),
>
> where $\mu$ and $\lambda$ are hyperparameters, $b^o_t$ is the amounts outgoing transfers from task $t$ to all other tasks, and $B \geq 0$ represents the set of all element-wise positivity constraints of a matrix.
>
> The following are the differences of our method from AMTL.
>
> - AMTL utilizes **full data** and does not perform any coreset selection, as it tackles MTL problems and thus does not face the problem of selecting the **smallest set of the most effective training instances** for rehearsal-based continual learning.
>
> - Moreover, since our algorithm is a coreset selection method, our objective is **different from, and orthogonal to**, that of AMTL which aims to learn to transfer knowledge across the task-specific predictors, and uses the **parameters** as the means for inter-task knowledge transfer.
>
> - Furthermore, while the main idea of AMTL is using the **task loss** to perform asymmetric knowledge transfer across tasks, our three coreset selection criteria, minibatch similarity, cross-batch diversity, and coreset affinity, **does not exploit task loss in any means** but focus more on building a representative and less redundant coreset for continual learning.
>
> However, we will include [Lee et al. 16] into the related work section as its motivation of dealing with task imbalance is related to one of the challenges we aim to tackle in online continual learning scenarios. We thank you for the suggestion.
>
> ---
>
> > **[3]** *Some grammatical errors and typos exist, such as 'let .. is' in definition 3.*
>
> $\rightarrow$ Thank you for finding this out, we have fixed the typos in the revision (updates are highlighted in blue).

---

> ### Author Response · Authors · 2021-11-21
> **A gentle reminder**
>
> Dear reviewer mR25,
>
> We sincerely appreciate your positive comments that our paper is well structured, contains sufficient experiments, and comes with meaningful and interesting motivation. We have made every effort to faithfully address all your comments in the responses.
> **We now have less than two days** to have interactive discussions. We have faithfully addressed your concerns regarding the difficulty of adapting existing algorithms to online learning and the misunderstanding on the novelty. The following is a quick summary of the response.
> ----
> - We have further clarified the difficulty of adapting existing algorithms to online learning. In the realistic CL scenarios, existing methods **suffer from sub-optimal convergence or biased training due to noisy and class-imbalanced data points** in the arriving data streams.
>
> - Moreover, to discern the advantage of our OCS in the online learning setting compared to baselines, we **did provide** the experimental results for the **online learning version of existing rehearsal-based methods** in **Table B.8** and **the first paragraph in Appendix Section B** in the original paper. This **highlights the clear contribution and effectiveness in our OCS** for online selection of helpful coresets during continual learning.
>
> - We also have clarified that our method is based on a **completely different motivation and idea from AMTL** [Lee et al. 16], except that we also consider imbalanced dataset scenarios, so that we have demonstrated a clear novelty and contributions of our OCS.
> ----
> We sincerely appreciate your insightful and constructive comments and thank you again for your time and efforts in reviewing our paper. Please let us know if you have any further questions.
>
> Best regards,
> Authors

---

> ### Author Response · Authors · 2021-11-27
> **Dear Reviewer mR25 - A Gentle Reminder**
>
> Dear reviewer mR25,
>
> We sincerely appreciate your efforts in reviewing our paper, and your constructive comments. We have responded to your comments and faithfully reflected them in the revision.
>
> As you know, now we have only a few days to have interactive discussions. Could you please go over our responses and the revision since end of the final discussion phase is approaching? Please let us know there is anything else we need to clarify or provide.
>
> Best,
> authors.

---

> ### Author Response · Authors · 2021-11-29
> **Dear Reviewer mR25 - The end of the discussion period ends today.**
>
> Dear reviewer mR25,
>
> **Today is the end of the discussion deadline.** Could you please go over our rebuttal and check the responses? We believe that we have addressed all your concerns and that including these discussions will further strengthen our paper. **We hope you reflect this in your final review and the score.** We thank you again for your time and efforts in reviewing our paper.
>
> Thank you,
> Authors.

---

### Decision · Program_Chairs · 2022-01-20

**Decision:**

Accept (Poster)

**Comment:**

The authors propose three strategies for coreset selection in the context of continual learning. In particular, the authors consider class-imbalance and noisy scenarios. The authors run extensive benchmarks and ablation showing that the approach can be effective in practice. All reviewers were positive about this work, but found that the methodological contributions were relatively modest. The clarifications provided by the authors were highly appreciated. I would encourage the authors to revise the paper to incorporate these additional details as there were a number of concepts that reviewers found were not sufficiently documented/explained and lacked clarity. I would also highly encourage the authors to explain their use of "online continual learning" as this reads like a tautology.

Finally, I would like to ask the authors to reflect on their insistance with the reviewers; while we would all want engaging and long discussions about our work, the reality is that reviewing papers and discussing them is time consuming and taxing, especially in the middle of continued pandemic. The authors should be grateful of the time reviewers have spent reading their work and providing feedback, and it is not in the authors' interest to ask for a revision of the scores.